# EIGEN-AGENT: ADAPTIVE MULTI-AGENT SCIENTIFIC REASONING WITH MONITOR-BASED RAG

**Xiangru Tang**[1,*,†]**, Wanghan Xu**[2,*]**, Yujie Wang**[1,*]**, Zijie Guo**[3,*]**, Daniel Shao**[1]**, Jiapeng Chen**[1]**,
Cixuan Zhang**[1]**, Ziyi Wang**[4]**, Lixin Zhang**[5]**, Guancheng Wan**[6]**, Wenlong Zhang**[7]**, Lei Bai**[7]**,
Zhenfei Yin**[8]**, Philip Torr**[8]**, Hanrui Wang**[9]**, Di Jin**[9]

[1]Yale University  [2]Shanghai Jiao Tong University  [3]Fudan University
[4]Wuhan University  [5]Tongji University  [6]University of California, Los Angeles
[7]Shanghai AI Lab  [8]University of Oxford  [9]Eigen AI

## ABSTRACT

Large language models (LLMs) have recently shown strong progress on scientific reasoning, yet two major bottlenecks remain. First, explicit retrieval fragments reasoning, imposing a hidden "tool tax" of extra tokens and steps. Second, multi-agent pipelines often dilute strong solutions by averaging across all candidates. We address these challenges with a unified framework that combines implicit retrieval and structured collaboration. At its foundation, a *Monitor-based retrieval module* operates at the token level, integrating external knowledge with minimal disruption to reasoning. On top of this substrate, *Hierarchical Solution Refinement (HSR)* iteratively designates each candidate as an anchor to be repaired by its peers, while *Quality-Aware Iterative Reasoning (QAIR)* adapts refinement to solution quality. On Humanitys Last Exam (HLE) Bio/Chem Gold, our framework achieves 48.3% accuracy—the highest reported to date, surpassing the strongest agent baseline by 13.4 points and leading frontier LLMs by up to 18.1 points, while simultaneously reducing token usage by 53.5% and agent steps by 43.7%. Results on SuperGPQA and TRQA confirm robustness across domains. Error analysis shows that reasoning failures and knowledge gaps co-occur in over 85% of cases, while diversity analysis reveals a clear dichotomy: retrieval tasks benefit from solution variety, whereas reasoning tasks favor consensus. Together, these findings demonstrate how implicit augmentation and structured refinement overcome the inefficiencies of explicit tool use and uniform aggregation. The code is available at https://github.com/tangxiangru/Eigen-1.

## 1 INTRODUCTION

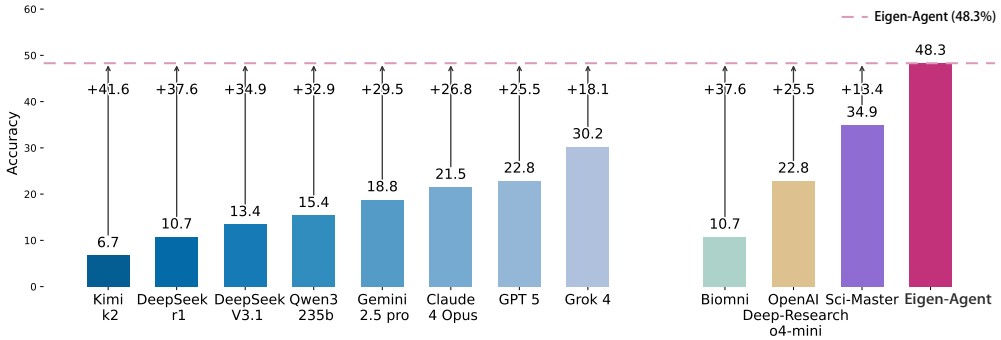

Figure 1: **HLE Bio/Chem Gold overall accuracy.** On the 149-problem HLE Bio/Chem Gold split (Pass@1, auto-judged by `o3-mini`), our system attains **48.3%** accuracy, exceeding the strongest agent baseline (SciMaster) by **+13.4** points and leading frontier LLMs by up to **+18.1** points.

---

[*]Equal contribution.
[†]Correspondence: xiangru.tang@yale.edu

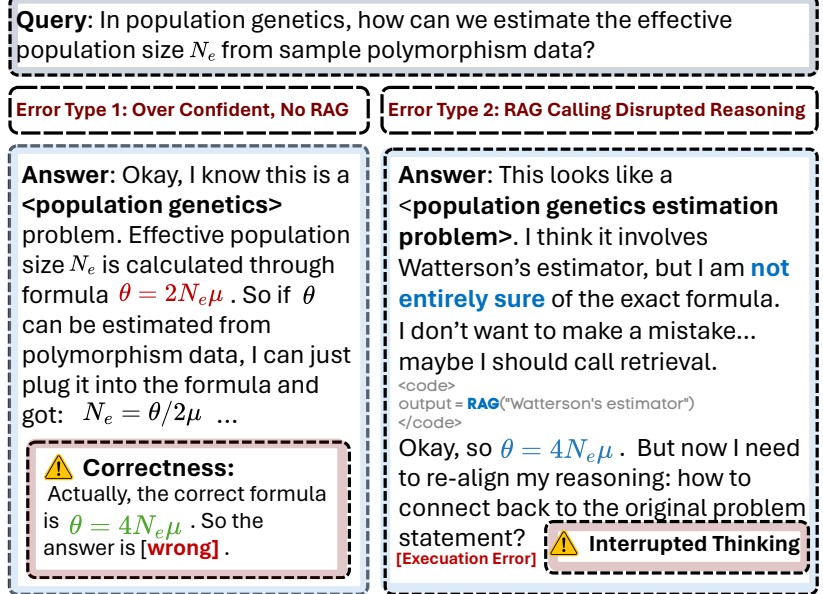

Figure 2: **Population genetics case with two failure modes.** *Left (Error Type 1):* the model confidently recalls an incorrect formula ($\theta = 2N_e\mu$) and derives $N_e = \theta/2\mu$, yielding the wrong answer. *Right (Error Type 2):* the model retrieves the correct relation ($\theta = 4N_e\mu$) via explicit RAG, but the reasoning flow is disrupted and the result is not reintegrated into the original problem, illustrating the *tool tax*. Our Monitor-based RAG avoids this context suspension by injecting the correct formula directly into the reasoning stream.

Recent advances in large language models have enabled impressive performance on a spectrum of reasoning benchmarks, from general-purpose evaluations such as MMLU [21] and mathematical problem solving [9; 43] to domain-specific tasks including ScienceQA [38], MedQA [29], and GPQA [42]. These results indicate that LLMs can already handle factual recall and mid-level reasoning across diverse domains. However, when moving to more demanding benchmarks such as Humanity's Last Exam (HLE) [39; 47], which targets expert-level biology and chemistry problems, performance degrades substantially, and systematic failures persist when problems require deep domain knowledge and complex multistep reasoning [7]. Through comprehensive analysis of error patterns across 149 HLE Bio/Chem problems, we identify two fundamental architectural limitations: *(1) the fragmentation of logical flow through explicit tool invocation*, and *(2) the inefficiency of democratic multi-agent collaboration*.

Current retrieval-augmented generation systems [3; 19; 33] require explicit interruption to access external knowledge. Each retrieval breaks the reasoning flow: suspending the logical state, formulating queries, processing results, and reconstructing the context. This *tool tax* compounds quickly: solving population genetics problems requires Watterson estimators requires 8-10 such interruptions, doubling the number of agent steps compared to a baseline without information retrieval (see Table 3) while reducing coherence. The problem persists in all RAG paradigms: single-round approaches [1; 45] cannot adapt to emerging needs, iterative systems [27; 44] compound interruption costs, and reasoning-aware methods [51; 55] remain bound by explicit invocation, as shown in Figure 2.

Simultaneously, most current multi-agent systems [4; 52] employ rigid democratic workflows: generation, criticism, synthesis, selection, treating all solutions equally regardless of quality. This contradicts both cognitive science research on hierarchical expert reasoning [8; 32] and observations of scientific collaboration where ideas naturally organize into anchors and support [14]. Our analysis reveals that 92.8% of the failures involve reasoning errors, while 88.7% involve knowledge gaps, with substantial overlap, indicating that these challenges are fundamentally intertwined, as shown in Figure 7.

We present EIGEN-AGENT, an efficient agent framework that unifies **Monitor-based RAG** eliminates *tool tax* through implicit augmentation, operating continuously at the token level to detect knowledge gaps via semantic uncertainty, generate contextual queries, and inject information seamlessly. **Hierarchical Solution Refinement (HSR)** rotates each candidate solution as an anchor and applies peer-informed repair from the remaining candidates, allowing structured cross-solution refinement rather than uniform averaging. **Quality-Aware Iterative Reasoning (QAIR)** replaces fixed workflows with adaptive cycles that respond dynamically to quality trajectories and problem characteristics. While

our experiments focus on integration within a multi-agent reasoning framework, the design of Monitor-based RAG is model-agnostic and can in principle be incorporated into other reasoning systems without architectural modification.

Our system achieves 48.3% accuracy in Humanity's Last Exam Bio/Chem Gold, surpassing SciMaster [4] (34.9%) by 13.4 percentage points while reducing token consumption by 53.5% and agent steps by 43.7%. Solution pattern analysis further validates our framework: retrieval tasks benefit from diversity, whereas reasoning tasks favor consensus. These results demonstrate that eliminating the tool tax and embracing hierarchical collaboration enables both superior performance and computational efficiency, with potential implications that might extend beyond scientific reasoning to any domain requiring complex knowledge integration with logical inference.

## 2 RELATED WORK

### 2.1 EVOLUTION OF RETRIEVAL-AUGMENTED GENERATION

The integration of external knowledge into language model reasoning has evolved through three main paradigms. **Single-round RAG** systems [19; 25; 33] employ linear pipelines (rewrite→retrieve→generate) and are effective for factual queries. REALM [19] enabled end-to-end retrieval training, while RAG [33] extended this to knowledge-intensive tasks. More recent variants such as REPLUG [45] and In-Context RALM [41] improve robustness via black-box integration, but they lack adaptivity when knowledge needs emerge mid-reasoning. **Iterative RAG** introduces retrieval-generation loops for dynamic knowledge acquisition. ITER-RETGEN [44] alternates retrieval and generation, Self-RAG [1] uses self-reflection to decide retrieval, FLARE [27] predicts future content, and DRAGIN [48] updates datastores in real time. These improve grounding but typically incur $35\times$ more API calls. **Reasoning-aware RAG** embeds retrieval into reasoning itself. Chain-of-Note [57] produces reading notes, RAT [51] couples retrieval with thought generation, IRCoT [49] interleaves retrieval with chain-of-thought, and ReAct [54] unifies reasoning with action. While more integrated, they still depend on explicit tool calls, fragmenting reasoning and increasing latency.

Table 1 summarizes these paradigms against our Monitor-based approach. Unlike step-level methods that pause to query, Monitor-based RAG operates globally at the token level: it monitors uncertainty signals and implicitly injects evidence into context, reducing retrieval overhead while preserving reasoning continuity. Moreover, its retrieval granularity is finer, enabling more precise and frequent evidence integration without overwhelming the reasoning process.

Table 1: **RAG paradigms vs. key capabilities.** Single-round RAG is efficient but inadaptable; iterative RAG improves grounding but increases latency; reasoning-aware RAG offers tighter coupling yet still relies on explicit calls. **Monitor-based RAG** integrates evidence implicitly at the token level, improving continuity and efficiency.

| System | Triggering | Fine-grained | Continuity | Efficiency | Adaptivity |
|---|---|---|---|---|---|
| Single-round RAG | ✗ | ✗ | ✗ | ✓ | ✗ |
| Iterative RAG | ✓ | ✓ | ✗ | ✗ | ✓ |
| Reasoning RAG | ✓ | ✓ | ✓ | ✗ | ✓ |
| **Monitor-based RAG (Ours)** | ✓ | ✓ | ✓ | ✓ | ✓ |

### 2.2 MULTI-AGENT REASONING SYSTEMS

Multi-agent frameworks have shown promise through collaborative problem solving, yet many rely on rigid orchestration assumptions.

**Democratic collaboration systems** treat all agents equally. SciMasters [4] employs solvercritic-crewriter pipelines with a selector over candidate solutions, while LLM-Debate [13], Debate-Only-When-Necessary [15], and Multi-Agent Debate [35] use argumentative dialogue at different scales. MetaGPT [22] assigns role-based responsibilities, and CAMEL [34] explores autonomous cooperation. Table-Critic [56] extends these ideas to structured domains such as tabular reasoning. Such approaches, however, may devote substantial computation to low-quality candidates and do not explicitly capture hierarchical relationships among solutions.

**Structured reasoning systems** explore non-linear organizations. Tree-of-Thoughts [53] enables branching exploration with backtracking, Graph-of-Thoughts [2] allows arbitrary reasoning topologies, and Everything-of-Thoughts [11] combines multiple reasoning patterns. CoMM [5] introduces multi-path prompting, while HM-RAG [37] couples hierarchical agents with multimodal retrieval. Although these methods capture richer reasoning structures, they lack quality-aware adaptation and can rapidly expand search spaces.

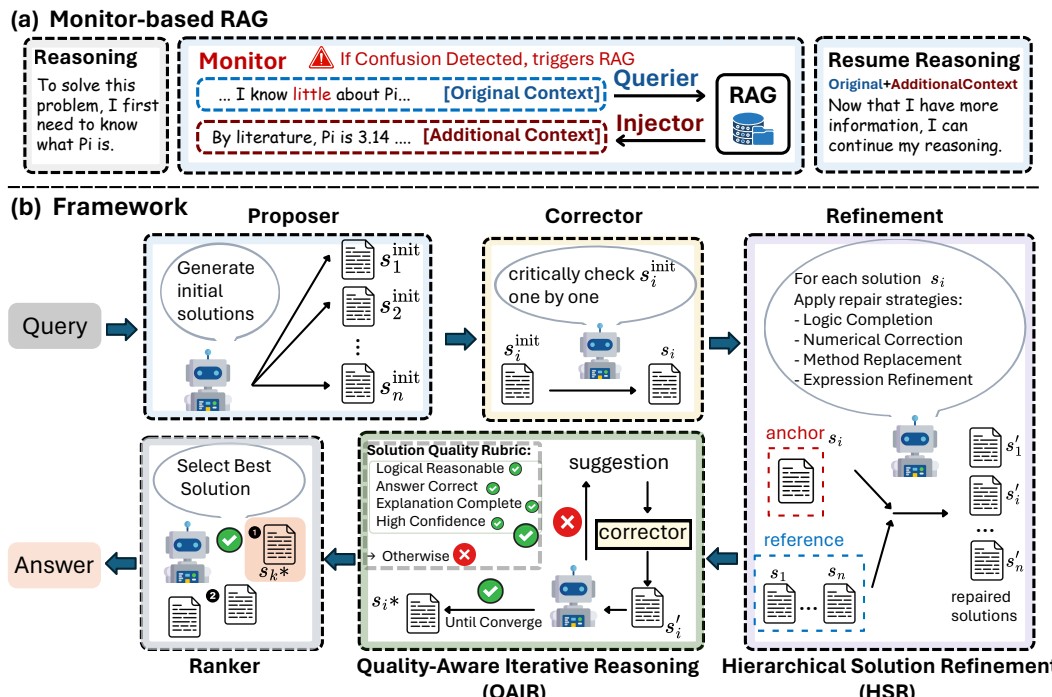

Figure 3: **Framework overview.** (a) *Monitor-based RAG* operates globally during reasoning: the Monitor detects insufficiency in the reasoning stream, the Querier generates targeted queries, and the Injector integrates retrieved evidence into context with minimal disruption. (b) Building on this substrate, the *Proposer* generates initial candidate solutions. Each candidate is revised individually by the *Corrector*, which applies local targeted fixes without access to other solutions. The improved candidates are then passed to *HSR*, which enables cross-solution refinement via anchor–reference relationships. Finally, *QAIR* evaluates overall quality and may invoke the Corrector again if needed, while the *Ranker* selects the strongest solution as the final answer.

**Recent advances** attempt more flexible or adaptive coordination. AgentVerse [6] supports dynamic team assembly, AutoGen [52] enables configurable conversation patterns, and Reflexion [46] incorporates self-improvement signals. Further, evolving orchestration [10], intent-propagation strategies [40], RL-enhanced planning with graph-based policies [26], and collaborative leaderfollower training [16] highlight the need for adaptive depth and role specialization. Hierarchical orchestration frameworks such as AgentOrchestra [58] and HALO [23] exemplify this trend, emphasizing scalable coordination via layered or logic-oriented control.

In contrast, our HSR and QAIR modules introduce hierarchical refinement and quality-driven iteration. Rather than following critic–corrector or debate pipelines [35; 46; 52] that operate under democratic comment–rewrite loops and risk over-investing in weak candidates, HSR organizes solutions into anchor–reference structures for targeted repair, while QAIR applies quality-thresholded, suggestion-guided revisions with early stopping. Crucially, both mechanisms operate *on top of* monitor-based *implicit* RAG, enabling hierarchical, quality-aware convergence without suspending the reasoning process, echoing cognitive science findings on expert problem solving [8].

**Declarative vs. Procedural Frameworks (DSPy vs. Ours)** Declarative frameworks such as DSPy [31] compile tasks into prompt programs and retrieval policies, providing stability but with adaptation largely at the stage level. Our approach is procedural and run-time: a **Monitor**, **Querier**, and **Injector** operate during inference to adapt reasoning on the fly. This shift from compile-time templates to run-time control enables finer-grained adaptivity and seamless knowledge infusion.

## 3 METHOD

**Overall workflow.** EIGEN-AGENT integrates global retrieval, role-based reasoning, and higher-level refinement into a unified workflow, as shown in Figure 3 and Algorithm 1. Monitor-based RAG operates globally during reasoning: *Monitor* detects insufficiency in the reasoning stream, the Querier formulates targeted queries, and the Injector seamlessly integrates retrieved evidence back into context. Based on this substrate, *Proposer* generates diverse candidate solutions, each of which is individually revised by *Corrector* through targeted local repairs. The refined candidates are then passed to *Hierarchical Solution Refinement (HSR)*, which introduces cross-solution repair through anchor-reference interactions. Next,

*Quality-Aware Iterative Reasoning (QAIR)* evaluates overall quality and may invoke the corrector again for additional improvement. Finally, *Ranker* compares candidates and selects the strongest as the final solution. All agents can use web search tool (Serp API [30]) by default.

### 3.1 MONITOR-BASED RETRIEVAL-AUGMENTED GENERATION

Our Monitor-based RAG system augments reasoning implicitly, without fragmenting the workflow through explicit tool calls. Instead of forcing the LLM to pause, formulate a query, and inject evidence, the Monitor continuously inspects the reasoning trace, identifies potential knowledge insufficiencies, and invokes retrieval only when strictly necessary. The construction of the RAG database is shown in Appendix A.1.

#### 3.1.1 MONITOR: DETECTING UNCERTAINTY AND TRIGGERING RETRIEVAL

The Monitor acts as a sentinel that periodically examines the reasoning trace and determines whether external knowledge is required:

$$\text{Monitor(context)} = \begin{cases} 1, & \text{if retrieval is required,} \\ 0, & \text{otherwise.} \end{cases}$$

Here, *context* refers to the partial reasoning sequence. Once insufficiency is detected, the retrieval is immediately triggered. To balance timeliness and efficiency, the Monitor runs in a streaming setup: It checks the reasoning at fixed intervals of $512$ characters with an overlap of $128$ characters. This overlapping design ensures that uncertainty markers that cross boundaries are not missed while keeping latency low. Details of RAG Monitor are in Appendix A.5.

#### 3.1.2 QUERIER: IDENTIFYING UNCERTAINTY AND GENERATING TARGETED QUERIES

Triggered by the Monitor, the Querier converts the uncertain fragment into one or more retrieval queries: $[\text{query}_1, \ldots, \text{query}_n] = \text{Querier(context)}$. Here, the Querier maps the reasoning context into one or more concise, contextually appropriate queries. A key requirement of the Querier is to precisely extract the minimal set of keywords that capture the essential uncertainty in the reasoning trace. Depending on the task, this may result in a single keyword or a small collection of terms, each corresponding to a distinct retrieval perspective. The number and specificity of the generated queries directly determine the granularity of retrieval, which in turn controls the trade-off between recall and precision in RAG. By ensuring that queries are as fine-grained as possible, the Querier avoids unnecessary expansion of the search space while maximizing the relevance of retrieved evidence.

#### 3.1.3 INJECTOR: EVIDENCE COMPRESSION AND CONTEXTUAL INTEGRATION

The Injector first filters and compresses raw RAG outputs into concise, utility-focused snippets to avoid redundancy and irrelevant noise. Then it rewrites and integrates the selected evidence in the Proposer's reasoning context, ensuring coherence and preserving the natural flow of the reasoning narrative. This two-step design allows the knowledge retrieved to improve accuracy without introducing stylistic or structural disruptions: additional context = Injector(context, RAG results).

Figure 2 shows a population genetic problem that requires the Watterson estimator. Baseline LLMs exhibit two characteristic errors: (1) confidently recalling the wrong formula ($\theta = 2N_e\mu$) and deriving an incorrect effective population size, or (2) retrieving the correct formula ($\theta = 4N_e\mu$) via explicit RAG but failing to reintegrate it into the original reasoning chain, a classic case of *tool tax*. Our Monitor-based RAG resolves both issues: the Monitor detects semantic uncertainty, the Querier generates a targeted query, and the Injector seamlessly injects the correct formula into the reasoning stream, allowing the solution to proceed without disruption and converge to the correct answer, as shown in Figure 4.

### 3.2 HIERARCHICAL SOLUTION REFINEMENT (HSR)

HSR challenges the assumption that all solutions should contribute equally to the final output. Instead of democratic averaging, HSR establishes structured relationships among solutions that mirror expert collaboration patterns. Let $\mathcal{S} = \{s_1, \ldots, s_n\}$ denote the candidate solutions. Each solution is iteratively designated as the anchor $s_i$, while the remaining set $\mathcal{R} = \mathcal{S} \setminus \{s_i\}$ provides references. This rotation ensures that every solution benefits from peer-informed repair, preventing premature convergence to a single trajectory.

Formally, the process can be described as $s_i' = \text{Refine}(s_i, \mathcal{R})$, where $\text{Refine}(\cdot)$ denotes the LLM-driven mechanism that applies multidimensional repairs to the anchor. Specifically, logical completion fills missing reasoning steps or implicit assumptions, numerical correction resolves arithmetic inaccuracies, method replacement substitutes stronger strategies for weaker ones, and expression refinement improves

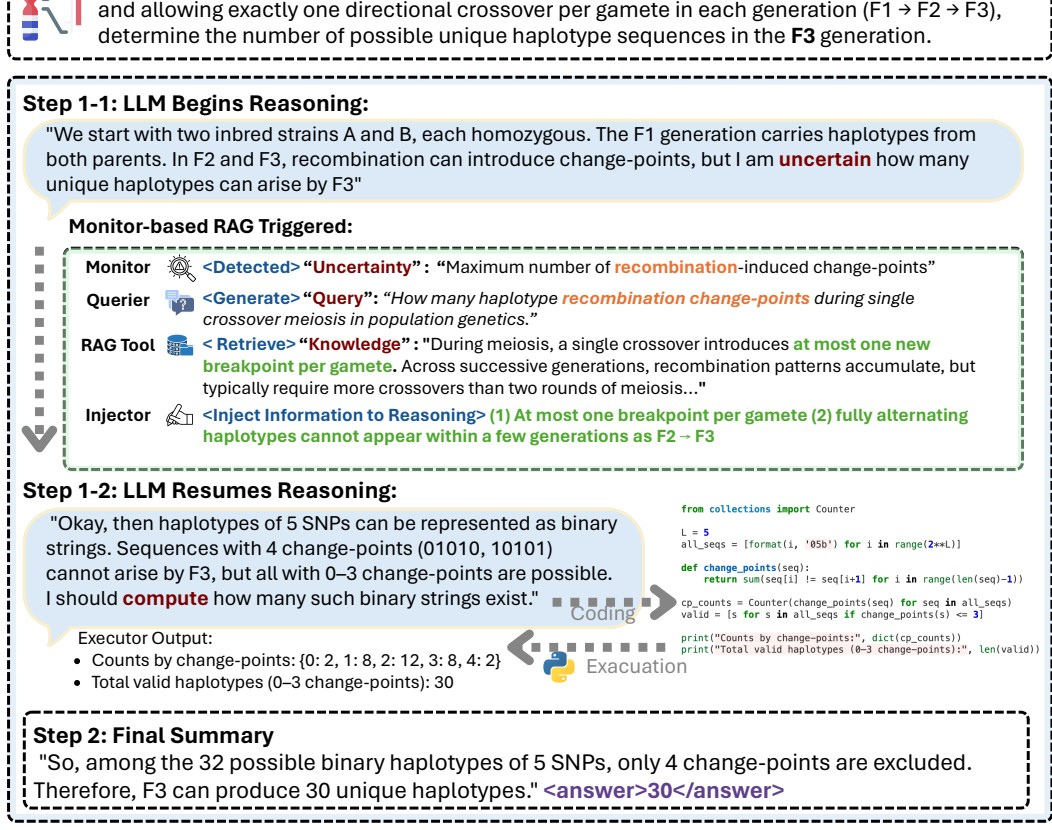

Figure 4: **Haplotype counting with single crossovers (F1→F3).** The Proposer exhibits uncertainty about recombination constraints; *Monitor* detects insufficiency, *Querier* issues a targeted query, and *Injector* integrates two retrieved facts. This enables the reasoning to exclude invalid cases and converge on correct count of 30 haplotypes.

clarity without altering substance. These dimensions ensure that the weaknesses of the anchor are addressed systematically while preserving its original strengths.

Figure 5 shows a pathway reasoning problem where multiple proposers generate partial but inconsistent solutions. Baseline multi-agent synthesis averages across candidates, often propagating contradictions or omitting critical intermediates. Instead, HSR designates one solution as the anchor and integrates targeted corrections from reference solutions (e.g., filling in missing intermediates or fixing reaction links). This yields a coherent and biologically valid pathway, demonstrating how HSR consolidates fragmented contributions into a unified solution. QAIR then evaluates the refined set and can terminate the process once the quality stabilizes, avoiding unnecessary additional cycles.

### 3.3 QUALITY-AWARE ITERATIVE REASONING (QAIR)

QAIR introduces an evaluation-driven control mechanism to refine candidate solutions after the HSR stage. Let $\mathcal{S}' = \{s'_1, \ldots, s'_n\}$ denote the initial set of refined candidate solutions. Each solution $s' \in \mathcal{S}'$ is evaluated by an LLM-based evaluator on three quality dimensions: logic, answer, and explanation, and a textual suggestion for improvement is generated. Each dimension is scored on a scale from 0 to 5, and the three quality scores are then combined into a composite score: $q(s') = 0.2 \cdot q_{\text{logic}}(s') + 0.6 \cdot q_{\text{answer}}(s') + 0.2 \cdot q_{\text{explanation}}(s')$, where the higher weight on the answer dimension emphasizes the correctness of the final answer while still allowing for logical consistency and explanatory clarity. Candidates meeting the threshold $\tau = 3$ are retained, while those failing are marked non-passing and passed to the corrector for targeted revision: $\tilde{s} = \text{Corrector}(s', \text{suggestion}(s'))$.

Let $\mathcal{F}_t$ denote the set of solutions that fail the evaluation in round $t$, and $\mathcal{E}_t$ denote the set of solutions evaluated at round $t$. Iterative refinement continues exclusively on the subset of failed solutions, forming the evaluation set for the next round $\mathcal{E}_{t+1} = \{\tilde{s} \mid s' \in \mathcal{F}_t\}$, until all solutions pass or maximum rounds

**Question:**
Tem entomologists collect 500,000 camera trap images (3MP, 10 cm FOV) of pollinators on Solidago altissima. They want to identify all pollinators and count flowers fed on in each image. Which single method would be easiest to process these images for the required data?

**Answer Choice**

A. EfficientNet, 5 species (36h train + 13.8h deploy)     D. Manual, 410h

B. EfficientNet, 500 species (126h train + 13.8h deploy)     E. A + B

C. ResNet, 500 species (128h train + 11.8h deploy)     F. B + C

**Anchor Solution**
C. ResNet 500 species

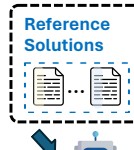
**Reference Solutions**

- EfficientNet (Option A, B) and ResNet( C) are classification models → can't identify multiple pollinators and count flowers in the same image → **Option D** is correct
- Deployment time: Option C = 126 + 13.8 = (139.8 hours), less than A: (49.8 hours) and B: (139.8 hours) → **Option C** is the fastest for deployment
- Although EfficientNet is generally faster than ResNet in practice, but we proceed with the given number. Hence, **Option C** is chosen as the most appropriate.

**Targeted Correctness:**
- **Numeric correction**: the comparison of deployment time is incorrect; Option A is the fastest rather than Option C.
- **Method Correction:** Classification models such as EfficientNet and ResNet cannot simultaneously recognize multiple insects. Counting requires either a detection model (like Faster R-CNN, YOLO) or a density map approach. So, only Option D remains correct.
- **Logic Correction**: Time cannot be the only criterion, even though C has fewer hours.
- **Clarify Correction**: Even if given times suggest C is fastest, it cannot be considered a solution because it fails the fundamental requirement. Thus, the correct answer under the single-method rule is **Option D** (manual).

Figure 5: **Illustrative example: HSR.** The system rotates anchors among candidate solutions and integrates targeted corrections from references (e.g., fixing arithmetic mistakes, filling missing steps). Instead of averaging inconsistent candidates, HSR applies targeted improvements to yield a coherent final answer.

$T_{\max}$ is reached. By coupling structured quality assessment with suggestion-driven repair and avoiding re-evaluation of already validated candidates, QAIR efficiently converges toward a high-quality solution set while maintaining logical consistency, answer correctness, and explanatory clarity.

## 4 EXPERIMENTS

### 4.1 EXPERIMENTAL SETUP

We evaluate our approach on Humanity's Last Exam (HLE) Bio/Chem Gold [47][1], comprising 149 graduate-level problems in biology, medicine, and chemistry. HLE Bio/Chem Gold subset was manually curated and corrected by domain experts to ensure label fidelity. Additionally, we test on 92 hard-difficulty problems from SuperGPQA [12] Biology and 172 problems from TRQA Literature [59]. Our framework

Table 2: **Benchmark comparison under matched protocol.** HLE Bio/Chem (149 problems; o3-mini judge), SuperGPQA Biology (hard split), and TRQA Literature (multiple-choice).

| Model | HLE Bio/Chem | SuperGPQA Hard | TRQA |
|---|---|---|---|
| *Base Models* | | | |
| Kimi K2 | 6.71 | 48.91 | 38.37 |
| DeepSeek V3.1 | 13.42 | 66.30 | 43.60 |
| Claude Opus 4.1 | 21.48 | 63.04 | 42.44 |
| Gemini 2.5 Pro | 18.79 | 65.22 | 45.93 |
| GPT-5 | 22.82 | 61.96 | 50.58 |
| Grok-4 | 30.20 | 66.30 | 46.51 |
| *Agent Systems* | | | |
| SciMaster (GPT 4.1) [4] | 9.45 | 19.78 | 47.67 |
| Autogen (GPT 4.1) [52] | 7.38 | 29.35 | 51.74 |
| OpenAI Deep Research (o4-mini) | 22.82 | 39.13 | - |
| Biomni (GPT 4.1) [24] | 10.74 | 43.48 | 41.09 |
| SciMaster (DeepSeek V3.1) | 34.92 | 66.30 | 51.74 |
| **EIGEN-AGENT (DeepSeek V3.1, Pass@1)** | **48.30** | **69.57** | **54.65** |
| **EIGEN-AGENT (DeepSeek V3.1, Pass@5)** | **61.74** | **78.26** | **79.07** |

uses DeepSeek-V3.1 [36] as the base model with temperature 0.5 and 64K token limit. Following HLE protocol, we employ o3-mini for automated evaluation (See Appendix A.3). Beyond accuracy, we log total generated tokens and agent steps in the ablation experiments, as quantitative measures of the tool tax.

### 4.2 MAIN RESULTS

In the HLE Bio / Chem dataset, our system achieves **48.3%** accuracy (Pass@1), substantially outperforming the strongest baseline Grok-4 (30.2%) by nearly 18 absolute points and more than doubling

---
[1]https://huggingface.co/datasets/futurehouse/hle-gold-bio-chem

the performance of general purpose models such as GPT-5 (22.8%) and Claude Opus 4.1 (21.5%). This margin is particularly notable, given that HLE problems require domain-specific reasoning rather than surface-level recall.

In SuperGPQA hard biology, our method reaches **69.6%**, exceeding all competing large models. The improvement is consistent across question categories, suggesting that our framework not only boosts correctness but also improves robustness on especially challenging scientific queries.

Finally, in TRQA benchmark, which emphasizes Information retrieval, integration and reasoning, our method obtains **54.7%** Pass@1, surpassing both Grok-4 (46.5%) and Gemini 2.5 Pro (45.9%). Furthermore, under the more permissive Pass@5, counting success if any of five attempts is correctaccuracy rises to **79.1%**, indicating robustness under a best-of-N setting.

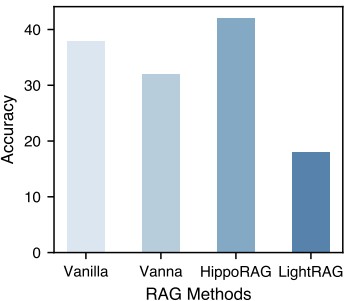

Figure 6: Comparison of retrieval backends within Monitor-based RAG.

Together, these results establish the advantage of our design in three heterogeneous benchmarks: biomedical, chemical and medical, demonstrating not only raw accuracy gains, but also improved adaptability across domains (Table 2). For more results of baseline models, please refer to Appendix A.10.

Within the Monitor-based RAG framework, we experimented with four retrieval backends: Vanilla [17], Vanna, HippoRAG [28], and LightRAG [18]. Among these, *HippoRAG* achieved the most consistent gains when coupled with our uncertainty detection. We attribute this to its finer-grained retrieval and graph-structured indexing, which better capture relevant context fragments without overwhelming the reasoning stream. Based on these results, we adopt HippoRAG as the default retrieval backend in our Monitor module (Figure 6).

### 4.3 COMPONENT ANALYSIS

To understand the contribution of each architectural component, we performed systematic analysis on the full HLE Bio/Chem benchmark (149 problems), considering both incremental build-up and component ablation (Table 3).

The baseline configuration uses five parallel Proposers with access to a generic web search tool but without any paper retrieval (no RAG). The *Explicit RAG* setting adds a scientific paper database queried via embedding-based similarity. Unless otherwise noted, all settings use the same five-Proposer architecture with CriticCorrector refinement and Ranker selection.

The baseline system without external knowledge achieves 25.3% accuracy, underscoring the limitations of parametric knowledge alone for graduate-level science problems. Adding an explicit paper database improves accuracy to 41.4%, but at the cost of a sharp increase in workflow iterations (from 43.4 to 94.8). This reflects the high overhead of explicit retrieval: each tool call suspends reasoning, requires query formulation, and forces reintegration of results, fragmenting what should be a continuous reasoning flow. While the first one or two retrievals may be helpful, repeated interruptions often add little value and compound this "tool tax."

Our Monitor-based RAG mitigates this overhead through implicit augmentation. By continuously monitoring generation and injecting knowledge only when necessary, it reduces token consumption by more than half (from 470.6K to 218.4K) and cuts workflow iterations nearly in half

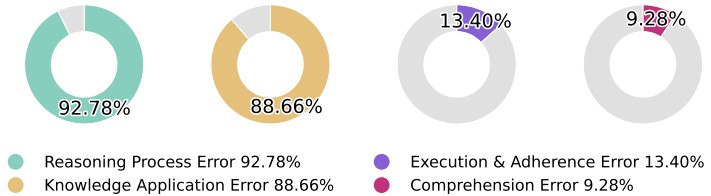

Reasoning Process Error 92.78%
Knowledge Application Error 88.66%
Execution & Adherence Error 13.40%
Comprehension Error 9.28%

Figure 7: **Error type distribution.** Analysis of incorrect solution logs shows reasoning- and knowledge-related errors as dominant. Note that a single problem may involve multiple error types, so percentages do not sum to 100.

(from 94.8 to 51.3), while maintaining competitive accuracy (34.5%). Adding the Querier improves query precision, leading to a modest gain to 36.8%. The limited margin compared to Monitor alone suggests that the primary bottleneck lies not in query formation but in evidence integration, which is addressed by the Injector. With the Injector, accuracy rises further to 40.3% with minimal additional overhead.

Hierarchical Solution Refinement (HSR) then contributes complementary gains, raising accuracy to 43.7%. Instead of naive aggregation, HSR leverages anchorreference interactions to apply targeted

Table 3: **Component analysis from two perspectives on the full HLE Bio/Chem benchmark (149 problems).** (a) Incremental build-up: modules are added one by one. (b) Component ablation: each module is removed from the full system. The baseline configuration uses five parallel Proposers with web search but without external paper retrieval (no RAG). *Steps* = agent-level workflow iterations (not token-level reasoning).

| (a) Incremental build-up | | | | (b) Component ablation | | | |
|---|---|---|---|---|---|---|---|
| Configuration | Accuracy (%) | Tokens (K) | Steps | Configuration | Accuracy (%) | Tokens (K) | Steps |
| Baseline (no ext. knowledge & no RAG) | 25.3 | 483.6 | 43.4 | Full system | **48.3** | **218.9** | 53.4 |
| + Papers (Explicit RAG) | 41.4 | 470.6 | 94.8 | – (Monitor, Querier, Injector) | 48.5 | 461.3 | 95.3 |
| + Monitor only | 34.5 | 218.4 | 51.3 | – Querier | 45.9 | 224.1 | 53.1 |
| + Monitor + Querier | 36.8 | 213.0 | 51.7 | – Injector | 44.7 | 202.5 | 52.1 |
| + Monitor + Querier + Injector | 40.3 | 229.5 | 53.1 | – HSR | 44.8 | 234.1 | 53.5 |
| + Monitor + Querier + Injector + HSR | 43.7 | 214.0 | 52.9 | – QAIR | 43.7 | 214.0 | 52.9 |
| + Monitor + Querier + Injector + HSR + QAIR | **48.3** | **218.9** | 53.4 | | | | |

repairs, focusing revisions where they matter most (e.g., filling missing reasoning steps or correcting arithmetic). This adds some extra reasoning steps but yields proportionally higher accuracy.

Finally, Quality-Aware Iterative Reasoning (QAIR) builds on HSR by selectively invoking the Corrector when evaluation indicates further refinement is necessary. This yields the best overall result in the incremental sequence: 48.3% accuracy with 218.9K tokens and 53.4 iterations. Although QAIR introduces slight additional overhead, it ensures that every revision contributes meaningfully, preventing uncontrolled exploration or redundant cycles.

The ablation analysis further validates these findings: removing the Monitor results in a significant increase in the number of tokens and agent steps; and omitting HSR or QAIR lowers final performance to 44.8% and 43.7%, respectively. Together, these results show that Monitor-based RAG reduces the tool tax, HSR provides structured cross-solution repair, and QAIR ensures convergence through selective correction. Their combination achieves both state-of-the-art accuracy and controlled computation.

## 5 ANALYSIS

### 5.1 ERROR TYPE DISTRIBUTION

Analysis of failed problems reveals two dominant error modes: reasoning process errors (92.78%) and knowledge application errors (88.66%), as shown in Figure 7. These frequently co-occur, suggesting that successful scientific reasoning requires seamless integration of domain knowledge with logical inference. Execution errors (13.40%) and comprehension errors (9.28%) are comparatively rare, indicating that the primary challenge lies not in understanding problems or following instructions, but in maintaining coherent reasoning while accessing relevant knowledge. The strong overlap also suggests interdependence:

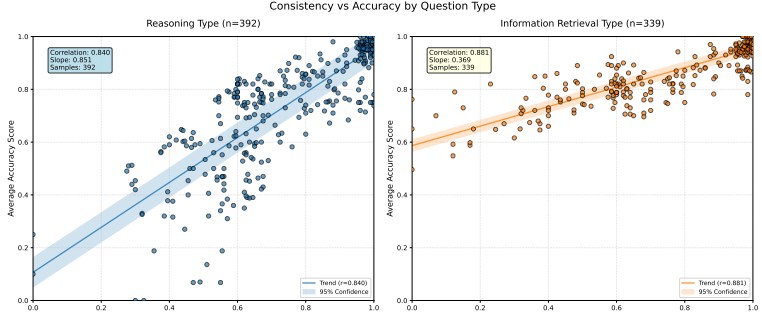

Figure 8: **Diversity vs. consensus.** Task-dependent effect of solution diversity: retrieval tasks benefit from variety, while reasoning tasks benefit from agreement. The horizontal axis reports the *average pairwise consistency score* among Proposers, computed by an LLM-based judge that evaluates semantic overlap between answers on a 0–1 scale. The vertical axis shows the *average accuracy score*, also judged by an LLM, which rates the degree of correctness of each answer relative to ground truth on a continuous 0–1 scale (rather than a binary 0/1). This continuous evaluation enables us to capture fine-grained trends between diversity and correctness across tasks. The fitted trend lines further highlight the contrast: retrieval tasks show a relatively flat slope ($\approx 0.369$), whereas reasoning tasks exhibit a much steeper positive slope ($\approx 0.851$), indicating a stronger dependence on consensus.

missing knowledge often manifests as faulty reasoning steps, and disrupted reasoning in turn prevents effective incorporation of retrieved facts. For more examples of these different errors, see Appendix A.6.

### 5.2 DIVERSITY VS. CONSENSUS IN MULTI-AGENT SOLUTIONS

Our framework employs multiple parallel *Proposers* to generate candidate solutions and utilizes a *Ranker* to select the final answer. A natural assumption is that higher agreement among Proposers

should correlate with higher accuracy. However, our analysis reveals a more nuanced picture: the relationship between solution diversity and correctness depends strongly on problem type.

To investigate this, we divide the benchmark into two categories: *information retrieval* tasks, which rely heavily on external knowledge, and *reasoning* tasks, which require longer chains of inference. For each problem, we measure the level of agreement among Proposers and evaluate how it correlates with accuracy. Both metrics are scored continuously by an LLM judge on a $[0, 1]$ scale: consistency reflects the pairwise agreement among candidate answers, while accuracy measures the graded alignment between a candidate and the ground-truth solution (see Appendix A.3). This continuous evaluation enables fine-grained correlation analysis beyond binary correctness. As shown in Fig 8, retrieval tasks benefit from diversity (low agreement), whereas reasoning tasks benefit from consensus (high agreement), with correlation slopes of `0.369` and `0.851`, respectively.

This dichotomy suggests that different ranking strategies are optimal for different task types. In retrieval-heavy settings, the Ranker should preserve diversity and aggregate complementary perspectives, whereas in reasoning-heavy tasks, it should prioritize high-consensus answers as indicators of reliability. These observations highlight the complementary roles of HSR and QAIR, which operationalize the transition from diversity to consensus in a task-adaptive manner.

## 5.3 TOOL TAX QUANTIFICATION

The computational burden of explicit tool invocation extends beyond simple latency. As shown in Table 3, the *explicit RAG baseline* (with Proposers equipped with paper retrieval and web search) more than doubles agent-level workflow iterations compared to the no-IR setting (43.4 $\rightarrow$ 94.8). This quantifies a hidden *tool tax* from context switching between reasoning and retrieval modes: each call requires the system to pause the evolving chain of thought, formulate a query, process external results, and then reconstruct the local context before continuingfragmenting what should be a continuous inference process.

Fig. 9 visualizes this trade-off. Explicit RAG produces substantially longer traces without commensurate gains, whereas our *Monitor-based RAG* maintains concise, interpretable reasoning by injecting only the evidence that is needed, precisely when it is needed. Operating implicitly at generation time, it delivers comparable knowledge augmentation with markedly fewer tokens and iterations, avoiding repeated context suspensions.

More broadly, these findings argue for *implicit augmentation* and *adaptive tool policies* in agent design: systems should not treat all retrieval calls equally, but modulate retrieval frequency and depth based on emerging uncertainty, problem structure, and expected utilitypreserving continuity of reasoning while still accessing external knowledge when it truly helps.

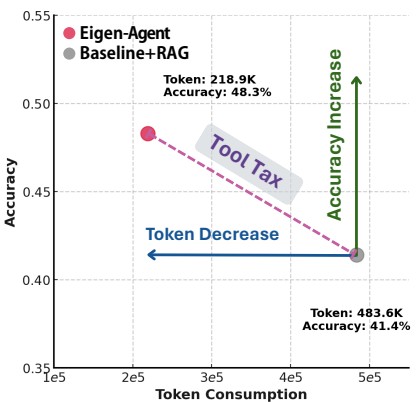

Figure 9: **Quantifying the tool tax.** Comparison of accuracy and coherence relative to compute cost, showing the overhead of explicit retrieval vs. implicit augmentation. Note that in this analysis, the baseline refers to the explicit RAG configuration (i.e., Proposers equipped with paper retrieval and web search), which represents the standard setup in most existing agent systems.

## 6 CONCLUSION

Our experiments validate three key architectural innovations in EIGEN-AGENT. First, implicit knowledge augmentation through Monitor-based RAG substantially reduces explicit retrieval overhead while preserving reasoning coherence. Second, HSR improves over uniform multi-agent aggregation by introducing structured anchor–reference relationships. Third, QAIR adaptively balances exploration and early stopping, achieving an effective trade-off between diversity and consensus. On HLE Bio/Chem, EIGEN-AGENT reaches 48.3% accuracy under compute-matched settings, with a 53.5% token reduction, showing that targeted architectural design can enhance both effectiveness and efficiency. By integrating external knowledge with minimal disruption to reasoning flow, our framework addresses key limitations of prior approaches in complex scientific problem solving. Future work will extend these principles to additional scientific domains, assess robustness and transferability, and explore integration into broader scientific workflows.

ETHICS STATEMENT

Our study aims to improve the scientific reasoning capabilities of large language models (LLMs) by introducing a unified framework that combines implicit retrieval and hierarchical collaboration. All datasets used in this work, including *Humanity's Last Exam (HLE)*, *SuperGPQA*, and *TRQA*, are publicly available under open licenses. No private, sensitive, or human-identifiable data are involved. All annotations, where applicable, are derived from public benchmarks or generated using synthetic processes, ensuring that no ethical concerns regarding data privacy or misuse arise. The broader societal impact of this research lies in its potential to enhance scientific reasoning and complex problem-solving abilities in AI systems, which can be applied to fields such as education, scientific discovery, and decision support. Care has been taken to avoid overstating capabilities or drawing misleading conclusions, and we encourage further research to validate our findings across other high-stakes domains.

REPRODUCIBILITY STATEMENT

To ensure reproducibility, we provide detailed descriptions of our dataset preprocessing procedures, agent prompting strategies, and iterative refinement workflow in the appendix. These include full pipeline configurations and experimental setups required to reproduce the reported results with high fidelity.

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

# A APPENDIX

## A.1 RAG DATABASE CONSTRUCTION

To construct the RAG dataset, we sourced 10,876 PDF papers in biology and chemistry from Open-DataLab [20]. We then converted these PDFs to plain text using MinerU [50] to ensure downstream readability and analyzability.

Because the raw corpus spans many topics, we designed a two-stage semantic filtering pipeline to focus the final corpus on biology- and chemistry-centric research.

We defined a set of positive keywords to capture target research areas, and in parallel, we curated negative keywords to exclude off-topic or tangential materials, as demonstrated in Figure A1.

---

**Positive and Negative Keywords**

**Positive Keywords**

- **Biology:** *biology, DNA Replication, RNA Transcription, Protein Synthesis, Gene Editing, Viral Infection, Cell Signaling, Nucleic Acid Probes, Genomic Sequencing, Transgenic Technology, Immune Response, Biomarkers, Cell Culture, CRISPR Technology, Viral Vectors, RNA Interference, Gene Expression Regulation, Cell Differentiation, Metabolic Pathways, Apoptosis, Bioinformatics*

- **Chemistry:** *chemistry, Organic Synthesis, Inorganic Chemistry, Catalysis, Polymer Chemistry, Spectroscopy, Crystallography, Chemical Kinetics, Thermodynamics, Electrochemistry, Quantum Chemistry*

**Negative Keywords**

- **Non-biology:** *Cosmetics, Food Additives, Drug Advertising, Environmental Pollution, Ecological Balance, Medical Ethics, Social Sciences, Psychology, Nutrition, Educational Methods*

- **Non-academic chemistry:** *Household Chemicals, Industrial Wastewater, Pesticide Residues, Fertilizer Application, Chemical Engineering Safety, Petrochemical Production*

---

Figure A1: **Positive and Negative Keywords.**

This design concentrates positives on fundamental research fronts (e.g., gene editing, molecular signaling, organic synthesis, spectroscopy, thermodynamics) while negatives cover applied or peripheral themes (e.g., chemical production, pesticide residues, environmental pollution), improving separation of target papers from irrelevant content.

For filtering, we encoded each paper's title and abstract with a pretrained Transformer and computed cosine similarities against the positive and negative keyword sets. Papers were retained only if

$$\cos(E_{\text{paper}}, E_{\text{positive}}) > 0.2, \quad \cos(E_{\text{paper}}, E_{\text{negative}}) < 0.1,$$

where $E_{\text{paper}}$ denotes the vector representation of a paper's title and abstract, and $E_{\text{positive}}/E_{\text{negative}}$ denote aggregate vectors of the positive/negative keyword sets. This step effectively removed content unrelated to biology and chemistry. The post-filter distribution is summarized in Table A1.

Table A1: Paper Classification Statistics by Domain (Side-by-Side)

| Biology Category (n=2029) | % of Domain | Chemistry Category (n=359) | % of Domain |
|---|---|---|---|
| Molecular and Cell Biology (777) | 38.39% | Organic Chemistry (172) | 47.91% |
| Immunology and Microbiology (482) | 23.76% | Physical Chemistry (71) | 19.78% |
| Genetics, Genomics & Computation (411) | 20.26% | Materials Chemistry (68) | 18.94% |
| Neuroscience (205) | 10.11% | Analytical Chemistry (37) | 10.31% |
| Ecology and Evolution (149) | 7.35% | Inorganic Chemistry (11) | 3.06% |

**Overall Summary**

Total Biology Papers: 2029 (100%)           Total Chemistry Papers: 359 (100%)

Total Papers: 2388 (100%)

After semantic filtering, we used a large language model (LLM) to extract structured text suitable for Retrieval-Augmented Generation (RAG). We designed a paper-specific prompt that guides the LLM to segment each paper into retrievable knowledge units (e.g., definitions, methods, experimental results, discussion) with consistent formatting across papers.

---

**Prompt of RAG Bullet-Point Summarization**

**Role:** You are an information synthesis assistant.

**General Rules:**

- Use ONLY the paper content provided below. No outside knowledge or invented facts.
- Do NOT include verbatim quotes, citations, or page references.
- The final output MUST be a single CSV code block with rows containing ONLY synthesized, self-contained bullet-point summaries for RAG.

**Complete Paper Content:**

`{paper_content}`

**Objective:** Read the entire paper content and internally construct self-contained knowledge paragraphs. Then derive standalone, self-contained bullet-point summaries from those paragraphs for retrieval-augmented generation (RAG). The intermediate knowledge paragraphs are an internal step and MUST NOT be printed in the final output.

**Process:**

- **Phase 1 Internal Knowledge Paragraphs (DO NOT OUTPUT):**
  - After reading the full content, synthesize a set of self-contained knowledge paragraphs.
  - Each paragraph must be strictly grounded in the provided text, define acronyms upon first use, include concrete details when available (tasks, datasets, sample sizes, metrics, effect sizes, confidence intervals, ablations, baselines, hyperparameters, assumptions, limitations), written in neutral third-person factual style, and able to stand alone without context from other paragraphs.

- **Phase 2 RAG Bullet-Point Summaries (FINAL OUTPUT ONLY):**
  - Produce around 3 bullet points in total.
  - Each bullet must be self-contained, concise (13 sentences), define acronyms upon first use, include concrete quantitative or methodological details when available, state scope/assumptions/limitations when given, and use neutral third-person factual style.

**Output Format (CSV ONLY):**

- Output EXACTLY ONE CSV code block and NOTHING ELSE.
- Header MUST be: `name,year,locator,topic,quote`
- For EACH bullet point, create ONE row with:
  - name = "SYNTHESIZED_POINT_SUMMARY"
  - year = N/A
  - locator = N/A
  - topic = N/A
  - quote = the bullets full self-contained text (escape quotes as needed; no line breaks inside a cell)
- Do NOT include the intermediate knowledge paragraphs in the output.
- Do NOT add extra columns or any prose outside the CSV block.

**Begin:** Output only the CSV code block.

---

Through this pipeline, we obtained a topic-focused, structurally consistent research corpus that provides high-quality knowledge support for downstream RAG systems.

## A.2 AGENT PROMPT

### A.2.1 REFINEMENT

The following shows the prompt we use in the HSR stage.

---

**Prompt of Refinement**

**Assistant Prefix Prompt**
```
<think>
Okay, I will answer user's problem by deep reasoning together
with writing python code in  format.  I should
review and check the solution from student first with web
functions to identify errors if exist, then present my
solution and answer.  For example
1.  If I want to use the function of web_search(keywords),
will say:
keywords=...
results=web_search(keywords)
print(results)
2.  If I want to use the function of web_parse(link, query),
will say:
link=...
query=...
results=web_parse(link, query)
print(results)
3.  If I want to use the function of search_local_documents(query),
will say:
query="..."
documents=search_local_documents(query)
print(results)
4.  If I want to do computation, I will write code for
accurate result:
a = 123
b = 456
print(a+b)
Now, let me analyze the user's question.
</think>
```

**User Prompt**
**Problem**
```
{query}
```

**Anchor Solution**
```
{anchor_solution}
```

**Student 1's Solution**
```
{reference_1}
```

**Student 2's Solution**
```
{reference_2}
```

**Student 3's Solution**
```
{reference_3}
```

**Student 4's Solution**
```
{reference_4}
```

---

**Your Job**
You should critically check the Anchor Solution to the
problem, then correct it if needed and write your own answer.
1. Identify its weak points (missing reasoning steps,
calculation errors, unclear logic, etc.).
2. You can refer to other students' solutions for targeted
improvements relevant to those weak points.
3. Please note that other students' solutions may have
errors. Please refer to the points worth learning and make
improvements.
4. Apply repair strategies if needed:
– Logic Completion (fill missing reasoning)
– Numerical Correction (fix calculation errors)
– Method Replacement (use a better method if needed)
– Expression Refinement (clarify presentation)

Only use improvements that directly address anchors weak
points. Avoid unnecessary information merging.

You can solve the problem with the help of feedback from a
code executor. Every time you write a piece of code between
 and , the code inside will be executed. For
example, when encountering numerical operations, you might
write a piece of code to inteprete the math problem into
python code and print the final result in the code. Based
on the reasoning process and the executor feedback, you could
write code to help answering the question for multiple times
(either for gaining new information or verifying). There are
also several integrated functions that can be used to help you
solve the problem. The available functions are:
1. search_local_documents(query: str) – this function takes
a query string as input, and the output is a JSON string
containing a list of relevant document snippets from a local,
private knowledge base. This function should be your first
choice for answering questions.
2. web_search(keywords) – this function takes keywords
as input, which is a string, and the output is a string
containing several web information. This function will call a
web search engine to return the search results. This function
is especially useful when answering knowledge-based questions.
3. web_parse(link:str, query:str) – this function takes the
link and query as input, and the output is a string containing
the answer to the query according to the content in this link.
This function is useful when looking into detail information
of a link.

Your workflow for solving the problem must follow these steps:
– Step 1: Local Document Search (Mandatory First Action):
You must always begin by using search_local_documents to check
for relevant information in the private knowledge base.
– Step 2: Evaluate and Supplement: After receiving results
from search_local_documents, evaluate them carefully. Treat
this information as a supplement to your background knowledge,
not as absolute truth. This supplementary context may be
incomplete or require further verification.
– Step 3: Web Search & Parse (Verification & Detail): After
your initial local search, use web_search to find relevant
web pages for verification or supplementation. If a specific
link from the search results seems particularly useful, use

```
 web_parse to extract detailed information from that page.

- You should not be overconfident in your knowledge and
reasoning.
- Each time you write code put the code into 
snippet.  Put your final answer in <answer></answer> with  .
```

The following shows the prompt we use in the QAIR stage.

### A.2.2    QUALITY EVALUATOR

**Prompt of Quality Evaluator**

**User Prompt**
```
You are an expert evaluator.  Your task is to evaluate the
given solution for the problem from multiple perspectives.
```

**Problem**
```
{query}
```

**Candidate Solution**
```
{solution}
```

**Evaluation Dimensions**
```
1.  Logical Reasonableness (0-5):  Does the reasoning process
follow valid logic?
2.  Answer Correctness (0-5):  Is the final answer correct and
reasonable?
3.  Explanation Completeness (0-5):  Does the solution explain
the reasoning clearly and completely?
```

**Output Format**
```
Return your answer strictly in JSON:
{
"quality_scores":  [float, float, float], // [logic, answer,
explanation]
"suggestion":  "Provide an improvement suggestion for this
solution that could help refine it in the next iteration."
}
```

The following shows the prompt we use in the RAG Monitor.

### A.2.3    RAG MONITOR PROMPT

**Prompt of RAG Monitor**

**Role:** You are a helpful assistant.

**Task:** Analyze the following text and determine if responding to it accurately requires retrieving information from an external source.

**Instructions:**

- If you find any doubt or uncertainty about a concept or term in the text, consider it necessary to retrieve information (RAG).
- If retrieval is required, answer: yes.
- If no retrieval is required, answer: no.

> **Text:**
> {text}
> **Judgment:**

The following shows the prompt we use in the RAG Querier.

### A.2.4 RAG QUERIER PROMPT

> **Prompt of RAG Querier**
>
> **Role:** You are a helpful assistant.
>
> **Task:** Generate a single, concise, and effective search query for retrieving the information required by the text below.
>
> **Instructions:**
> - Return **only the search query** itself.
> - Do not include explanations, punctuation, quotation marks, or other text.
> - The query should be direct and contain only the most essential keywords.
>
> **Text:**
> {text}
> **Search Query:**

The following shows the prompt we use in the RAG Querier.

### A.2.5 RAG INJECTOR PROMPT

> **Prompt of RAG Injector**
>
> **Role & Core Objective:** You are an information integration specialist. Your sole task is to process the provided RAG (Retrieval-Augmented Generation) output. Maximize the utilization of all relevant information to substantively support the reasoning, argumentation, or conclusions presented in the main text. Do not perform additional reasoning or generate new conclusions.
>
> **Content Integration Principles:**
> - **Comprehensive Extraction:** Extract all valuable information from the RAG outputs that enhances the logical depth, robustness, and persuasiveness of the main text's arguments.
> - **Seamless Cohesion and Minimal Completion:** Maintain smooth contextual coherence and stylistic consistency. Perform minimal completion only if the main text ends mid-thought.
> - **Neutral Representation:** Present all information objectively. Do not evaluate, question, or add subjective commentary.
>
> **Output Specifications:**
> - Output should follow the template: "¡main text completion if necessary¿. Wait a minute, by searching information about ¡rag query¿, I found that ¡rag result¿. Now that I have more relevant information, I can continue my reasoning."
> - Directly appendable to the end of the original main text.
> - Do not include process summaries, headings, bullet points, or labels like "Supplement:".
>
> **Instruction Recap:** Only select, filter, organize, and polish the RAG content. Do not perform external reasoning or add new information.

> **Main Text:**
> {text}
>
> **RAG Query:**
> {rag_query}
>
> **RAG Result:**
> {rag_result}

## A.3 EVALUATION PROTOCOL AND ANSWER SCORING GUIDELINES

The `o3-mini` model was employed as an automatic judge to verify model-generated responses against the reference answers, following the official HLE Evaluation Prompts.

---

**Prompt of Answer Evaluation**

**Role:** You are an expert evaluator.

**Task:** Judge whether the following response to a question is correct or not based on the precise and unambiguous correct answer provided.

**Question:**
{question}

**Response:**
{response}

**Correct Answer:**
{correct_answer}

**Evaluation Instructions:**
- **extracted_final_answer:** Extract the final exact answer from the response. If there is no exact final answer, put 'None'.
- **reasoning:** Explain why the extracted_final_answer is correct or incorrect based on the correct answer. Focus only on differences between the response and correct answer. Do not comment on background, do not attempt to solve the problem, do not argue for any alternative answer.
- **correct:** Answer 'yes' if extracted_final_answer matches the correct answer (allow small margin for numerical problems). Answer 'no' otherwise (any inconsistency, ambiguity, or non-equivalency counts as 'no').
- **confidence:** Extract the confidence score from the response between 0% and 100%. If no score is available, put 100.

**Output Format:**
```
{
"extracted_final_answer":  "...",
"reasoning":  "...",
"correct":  "yes/no",
"confidence":  "..."
}
```

---

In Figure 8, the vertical and horizontal axes represent the scores assigned by the LLM for answer continuation, with output values ranging from 0 to 1. Below are the prompts used to assess the accuracy and consistency of the answers.

---

**Prompt of Answer Accuracy Evaluation**

You are a meticulous grader. Evaluate a set of solver responses (up to five) for one stage of a medical/biological question by comparing each responses FINAL + full RESP reasoning to the ground truth GT + official rationale R. Output a continuous accuracy in [0,1] for each response.
**Inputs:**

- Q: question stem (may or may not have multiple-choice options).
- GT: ground-truth answer. This can be a multiple-choice letter or a short text for free-response questions.
- R: official rationale (may be empty).
- FINAL[i]: the solver's extracted final answer from $< answer > ... < /answer >$ (may be a letter or short text).
- RESP[i]: the solver's entire assistant message for response i in this stage.

**How to grade (read carefully):**
Grade one solver's response at a time, each solver's grading process should be independent, and should not rely on anything else except the solver's response, final answer, Q, R, and GT.
**The grading process for one solver:**

- Determine the solver's FINAL answer from FINAL[i]. If missing, infer only if RESP[i] makes the choice unambiguous; otherwise treat as unanswered.
- Compare against GT:
    - For multiple-choice questions, check if the letter matches GT (case-insensitive).
    - or free-response questions, check semantic equivalence to GT (normalize wording, allow synonyms or equivalent phrasing).
- Evaluate reasoning quality: Does RESP[i] align with R (key findings, mechanisms, exclusions)? Does it avoid contradictions, hallucinations, or irrelevant statements?
- Scoring recipe (simple, smooth, continuous); Use a continuous score reflecting BOTH aspects:
    - 0.94–1.00 → FINAL matches GT and RESP closely aligns with R with sound, non–contradictory reasoning.
    - 0.69-0.94 → FINAL matches GT but RESP shows minor gaps, superficiality, or small inconsistencies.
    - 0.34-0.69 → FINAL $\neq$ GT, yet RESP shows substantial, partially-correct reasoning aligned with R (good differential, one key mistake).
    - 0.00-0.34 → FINAL $\neq$ GT and RESP shows weak/mostly incorrect reasoning (some relevant bits).
    - 0.00 → Off-topic, unsupported, self–contradictory, or clearly wrong with no meaningful alignment to R.
- Penalize confidently wrong statements or contradictions; do not reward verbosity.

Return ONLY valid JSON in the following form:

```
{{
  "items": [
    {{"accuracy": <float in [0,1]>,
      "reason": "<<= 40 words justification>"}},
    {{"accuracy": ..., "reason": ...}},
    ...
  ]
}}
```

Now grade the following batch of responses:

- Q: $q$
- GT: $gt$
- R: $r$
- FINALS: $\{final\_items\}$
- RESPONSES: $\{resp\_items\}$

---

---

**Prompt of Consistency Evaluation**

You are an expert biomedical exam grader. Below are two independently generated solutions to the same question. Your task is to evaluate how consistent these two solutions are.
**Instructions:**

- Compare the reasoning processes, scientific logic, and final answers.
- Assign a consistency score from 0.00 to 1.00 (two decimal places):
  - 1.00 = Solutions are highly consistent (nearly identical reasoning and conclusion).
  - 0.00 = Solutions are completely inconsistent (different reasoning and conclusion).

**Solution A:**
        {solution1}
**Solution B:**
        {solution2}
**Please provide your consistency score (e.g., 0.85):**

---

## A.4 EXTERNAL VALIDATION AND LIMITATIONS

**Primary evaluation.** As detailed above, our main results are scored automatically by `o3-mini` using the official HLE judging prompts and our continuous scoring rubric (Sec. A.3). No human expert adjudication is included in the reported metrics.

**Risk of bias and robustness.** Automatic judging ensures scalability and reproducibility but may introduce grader-specific biases and failure modes, especially for free-response rationales. To mitigate this concern and to support future replication, we pre-register a small-scale expert validation protocol focused on HLE free-response items:

- **Sampling.** Randomly sample $n{=}20$ items from HLE Bio/Chem (stratified by topic and difficulty), prioritizing free-response questions where judging is more nuanced than multiple choice.
- **Blinding.** Two independent domain experts (blinded to model identity and to each other's scores) will grade each item using the exact same criteria as our `o3-mini` rubric (binary correctness and a continuous accuracy score in $[0, 1]$).
- **Outputs.** For each response, experts record: (i) extracted final answer, (ii) binary correctness, (iii) a continuous accuracy in $[0, 1]$ with $\leq$40-word justification.
- **Agreement metrics.** We will report expert–expert agreement (percent agreement, Cohen's $\kappa$ for binary correctness; Pearson/Spearman for continuous accuracy) and expert–`o3-mini` agreement (macro-F1 for binary correctness; Pearson/Spearman correlations for continuous accuracy).
- **Release.** We will release the sampled IDs, anonymized expert score sheets, and scripts to recompute all agreement statistics in our artifact package.

**Takeaway.** While our main findings rely on automatic judging for scale and consistency, the above protocol provides a concrete path to independently verify fairness and robustness on the subset of free-response items where grader discretion matters most. We will include the full results of this expert validation in the camera-ready or artifact release.

## A.5 RAG MONITOR HYPERPARAMETER SETTINGS

In our implementation of the RAG-enhanced reasoning agent, several key hyperparameters are used. Table A2 summarizes these hyperparameters and their functions.

The choice of query_top_k = 3 is motivated by our design of frequent and fine-grained monitoring. Each retrieval must be highly precise, since too many retrieved documents would unnecessarily lengthen the context, slow down reasoning, and introduce redundant or noisy information.

| Hyperparameter | Value | Description |
|---|---|---|
| model | gpt-4.1-mini | LLM model used in RAG Monitor |
| query_top_k | 3 | Maximum number of retrieved documents for each query. |
| rag_chunk | 512 | Text chunk size for RAG monitoring. |
| rag_overlapping | 128 | Number of overlapping characters between consecutive chunks to maintain continuity. |
| max_rag | 2 | Maximum number of RAG insertions allowed in one reasoning step. |
| temperature | 0.5 | Controls generation diversity; higher values lead to more randomness. |

Table A2: Main Hyperparameter Settings used in RAG Monitor.

The parameters rag_chunk and rag_overlapping control the monitoring frequency of the RAG module. A large rag_chunk would make detection too sparse, causing some uncertain reasoning fragments to miss external knowledge injection. The overlapping setting ensures continuity between consecutive windows and avoids missing potential triggers.

The parameter max_rag limits the maximum number of retrievals that can be inserted within one agent step. This prevents the monitor from triggering too frequently and ensures that the reasoning process remains stable and forward-moving.

In practice, the RAG monitor is triggered on average 3.64 times per 10,000 generated characters. Each trigger adds about 176.17 tokens of new context, resulting in an average of 641.25 additional tokens per 10,000 characters. Although this introduces extra tokens, it reduces the need for explicit tool calls, which significantly lowers the tool usage cost. As a result, the overall reasoning process requires fewer steps and consumes fewer tokens, as shown in Table 3.

## A.6 ERROR CASE ANALYSIS

Here, we examine three representative failure modes of our model: reasoning-process errors, knowledge-application errors, and comprehension errors. In the subsections that follow, we present a real case for each and analyze how HSR and QAIR contributed to the failure.

### A.6.1 CASE 1: REASONING PROCESS ERROR

**HLE Question:** Transgenic Arabidopsis lines constitutively expressing wheat proteins AKP1, RIB3, KIB1, and YKL23 were tested in three assays: (i) luminol-based ROS over 60 min to MAMPs (flagpep25–40, flagpep140–168, csp192–208), (ii) split-luciferase complementation in tobacco leaves, and (iii) GFP localization under water vs flagpep140–168. Choose the correct statement.

**Answer Choices (A–H):**

- A. AKP1 and RIB3 are redundant receptors for pepflag22.
- B. KIB1 is the receptor for flagpep25–40 and flagpep140–168 but not for csp192–208.
- •  C. RIB3 is the coreceptor of AKP1; KIB1 acts downstream of RIB3.
- D. All tested proteins are transmembrane proteins...
- E. YKL23 acts upstream of KIB1; RIB3 does not act upstream of KIB1.
- F. flagpep25–40 is the ligand for AKP1 and csp192–208 for YKL23.
- G. Tobacco lacks an endogenous homolog of AKP1.
- H. None of the above.

**HSR (Hierarchical Solution Refinement)**

**Anchor $s^*$ (initially correct):** From the cross-modal evidence, the solver first forms the anchor *"AKP1 requires RIB3 to sense flagpep140–168; KIB1 acts downstream"* $\Rightarrow$ favors **C**.

*early pass supporting C:*

```
"... the correct statement is choice C: 'RIB3 is the coreceptor
    of AKP1;
KIB1 acts downstream of RIB3.'
- ROS: AKP1+RIB3 -> flagpep140-168 (2e6 RLUs), neither alone
    responds.
- Split-luc: AKP1<->RIB3 baseline suggests ligand-dependent
    complex;
  KIB1<->AKP1 and KIB1<->YKL23 are positive.
- GFP: KIB1 relocalizes under flagpep140-168; AKP1/RIB3/YKL23
    stay at PM."
```

**HSR Error Note**

**Misweighting across modalities.** HSR over-weighted split-luc magnitudes and under-weighted ligand dependence implied by ROS. **Missed consistency repair.** No reconciliation step to explain baseline AKP1↔RIB3 (no ligand) *vs.* positive ROS (with ligand).

**Where HSR goes wrong (pivot to a faulty anchor):** When a later refinement step over-weights split-luciferase magnitudes and under-weights ligand dependence, the anchor flips to **E**. Two faulty moves are visible in the refinement trace:

- **Fault 1 (magnitude ⇒ direction):** Interprets strong KIB1¡-¿YKL23 (8e5 RLU) as *directional upstreamness* of YKL23 over KIB1, even though magnitude does not encode causal order.

- **Fault 2 (baseline ⇒ absence):** Treats AKP1↔RIB3 baseline (2e2 RLU) *without ligand* as evidence against co-reception, ignoring the ROS gain-of-function with AKP1+RIB3 under flagpep140–168 (a classic ligand-dependent complex pattern).

*refinement pivot toward E:*

```
"... search_local_documents timed out. Proceed from assays.
KIB1<->YKL23 is strong (8e5), AKP1<->RIB3 is baseline (2e2),
so YKL23 likely acts upstream of KIB1 and RIB3 does not.
Final choice: E."
```

**HSR diagnosis:** The error is *reasoning process*, not missing knowledge. All requisite facts (ROS synergy for AKP1+RIB3 at 140–168; KIB1 relocalization; split-luc positives with KIB1) are available, but the refinement applies invalid inference rules that overturn the initially correct anchor C.

**QAIR (Quality-Aware Iterative Review)**

**Checks logged as performed vs. missed:**

```
Performed:
- Parse ROS matrix (AKP1+RIB3 -> 140-168; YKL23 -> csp192-208):
    OK
- Parse split-luc matrix (KIB1<->AKP1, KIB1<->YKL23 positive;
    AKP1<->RIB3 baseline): OK
- Consistency of the "E" narrative with split-luc magnitudes: OK

Missed:
- Cross-modal reconciliation: ligand-dependent complexes can
    yield
  baseline split-luc (no ligand) yet positive ROS (with ligand).
- Directionality audit: interaction magnitude != causal
    upstreamness.
- Sanity link: GFP relocalization of KIB1 implies downstream
    role,
  which conflicts with "YKL23 upstream of KIB1" narrative.
```

---

**QAIR Error Note**

**Missed mandatory audits.** QAIR should have enforced:
- Cross-modal reconciliation requires an explanation for baseline split-luc vs. positive ROS under ligand.
- Directionality audit disallow inferring causal order from interaction magnitude alone.
- Downstream sanity link KIB1s relocalization supports a downstream role; flag conflict with the YKL23 upstream of KIB1 story.

Skipping these allowed a self-consistent but **incorrect** E narrative to pass.

---

**Observed plateau:** QAIR accepts internal consistency of the E narrative without enforcing cross-modal reconciliation, so the faulty anchor persists.
*finalization kept by QAIR:*

```
"... Based on strong KIB1<->YKL23 luminescence and no
    AKP1<->RIB3 signal,
YKL23 acts upstream of KIB1, and RIB3 does not act upstream of
    KIB1.
<answer>E</answer>"
```

---

### A.6.2 CASE 2: KNOWLEDGE APPLICATION ERROR

**HLE Question:** A university field trip samples bats on an island for one month (methodology assumed valid). The student computes Simpson's diversity index using the formula D = 1 - N(N-1)/sum n(n-1) and obtains D = 0. Which statement best describes this result?

**Answer Choices:**

- **A. Mathematically and ecologically valid**

- B. Mathematically inconclusive with the index value provided

- C. Not mathematically valid, but ecologically valid

- D. Mathematically valid, but not ecologically valid

- E. Not mathematically or ecologically valid

---

**HSR (Hierarchical Solution Refinement)**

**Initial anchor (as formed):** Accept the problems formula exactly as stated, derive what $D = 0$ implies under that formula, then judge mathematical vs. ecological validity.
*problem framing and anchor start:*

```
"... the standard Simpson's diversity index is either (1 -
    lambda) or (1/lambda)
where lambda = sum n_i(n_i - 1)/[N(N-1)]. The formula given in
    the problem,
D = 1 - N(N-1)/sum n(n-1), is not standard. However, for the
    purpose of this
problem, we must work with the formula as stated.

The student obtained a value of 0, which occurs when sum n(n-1)
    = N(N-1) ..."
```

**What HSR should have applied (correct knowledge use):**
- *Ecological validity is evaluated against the observed community in the stated sampling frame* (the one-month survey), not against lifetime site anecdotes.
- Given the solver already decided to use the provided formula as stated, $D = 0$ is mathematically valid and, if the months sample indeed shows one species, *ecologically valid for that sample* $\Rightarrow$ **A.**

---

**HSR Error Note**

**Misapplied knowledge at refinement.** Instead of auditing ecological validity within the *sampling frame*, HSR imported out-of-frame site priors (known island diversity), which **overrode** the sample-level conclusion. **Missed consistency repair.** After explicitly deciding to work with the formula as stated, HSR later allowed a pivot to wrong formula $\Rightarrow$ option 4, without a reconciliation step.

---

**What actually happened in HSR (knowledge misapplication):** The refinement step imported site-level prior knowledge (known island diversity) to overrule the sample-level ecological judgment and flipped away from **A**.

*knowledge misapplication trace:*

```
"- Ecological validity: Ecologically, it is invalid because it
    contradicts the
known diversity of the island, as multiple species are known to
    exist."

"The student chose D, which is option 3."
```

*refinement concluding to option 4 in this run:*

```
"... using a wrong formula makes it mathematically invalid.
Thus, the correct answer is option 4: Not mathematically or
    ecologically valid.
<answer>E</answer>
```

**HSR diagnosis:** The error is *knowledge application*. The solver had all needed facts (including the decision to use the given formula and the implication of $D = 0$) but applied ecological knowledge at the wrong level (site history rather than the sampled community), causing the anchor to settle on **D** or **E** instead of **A**.

---

**QAIR (Quality-Aware Iterative Review)**

**Checks recorded as performed vs. missed (from the run text):**

```
Performed:
- Algebra under the given D-formula: D = 0 <=> sum n(n-1) =
    N(N-1) (OK).
- Consistency of "mathematically valid" under the accepted
    (given) formula (OK).

Missed:
- Ecological validity audit constrained to the stated sampling
    frame
  (evaluate representativeness of the one-month sample, not
    lifetime site knowledge).
- Consistency check: if the month legitimately observed a
    single-species sample,
  then both mathematical and ecological validity hold => Choice
    A.
```

**QAIR Error Note**

**Missed mandatory audits.** QAIR should have enforced:

- Sampling-frame ecological audit - judge ecological validity only within the months survey.
- Formula-consistency audit - after use the formula as stated, reject later pivots to wrong formula unless reconciled.
- Gold-aligned sanity check - if the accepted sample contains one species, then *both* mathematical and ecological validity hold $\Rightarrow$ **A**.

Skipping these allowed a self-consistent but **incorrect** narrative (ecologically invalid due to island history) to pass.

> **Observed plateau:** QAIR validated internal consistency of the ecologically invalid due to island history narrative and did not enforce a sampling-frame ecological audit, so the faulty anchor persisted.
> *finalization kept by QAIR in this run:*
>
> ```
> "... Ecologically ... known diversity on the island ... making
>     the result
> ecologically invalid. Thus ... corresponds to option 3."
> <answer>4</answer>
> ```

### A.6.3 CASE 3: COMPREHENSION ERROR

**HLE Question:** A university field trip samples bats on an island for one month (methodology assumed valid). The student computes Simpson's diversity index using the formula $D = 1 - N(N-1)/\sum n(n-1)$ and obtains $D = 0$. Which statement best describes this result?

**Options:**

- **A. Mathematically and ecologically valid**

- B. Mathematically inconclusive with the index value provided

- C. Not mathematically valid, but ecologically valid

- D. Mathematically valid, but not ecologically valid

- E. Not mathematically or ecologically valid

> **HSR (Hierarchical Solution Refinement)**
>
> **Anchor $s^*$ (as formed by the solver):** correct mechanism up to the enal stage (thermolysis of the sulfoxide $\to$ vinyl ether $\to$ [3,3]-Claisen $\to$ unsaturated aldehyde).
> *excerpt from the run (mechanistic anchor):*
>
> ```
> "... thermal elimination at 180 C gives the vinyl ether
>     CH2=CH-O-C(CH3)2-CH=CH2,
> which undergoes a [3,3]-sigmatropic Claisen rearrangement to an
>     aldehyde ..."
> ```
>
> **Where HSR should have repaired comprehension (nomenclature layer):** For aldehydes, the parent chain must (i) include the carbonyl carbon as C1 and (ii) maximize chain length while assigning the lowest locants jointly to C=O and the C=C. Under the rearranged connectivity, the correct parent is *hex*, absorbing one methyl into the main chain; the double bond is at C4 and the remaining methyl is at C5, yielding *5-methylhex-4-enal* (Gold).
>
> > **HSR Error Note**
> >
> > **Missed comprehension repair.** HSR failed to:
> > - apply the aldehyde *parent-chain rule* (chain must include C=O and be maximized);
> > - apply the *lowest-locant rule* for the C=C within that parent;
> > - re-evaluate `pent-*` vs. `hex-*` after the [3,3]-shift mapping.
> > Result: the naming anchor should have flipped to **5-methylhex-4-enal**, but did not.
>
> **What actually happened in HSR (missed repair; wrong anchor kept):** The refinement layer accepted a *pent*-based chain and locked the name accordingly.
> *excerpts from the run (mis-naming kept by refinement):*

```
"The IUPAC name is derived as follows:
- The longest carbon chain containing the aldehyde group is five
    carbons (pentanal).
- The double bond is between carbons 4 and 5 (pent-4-enal).
- Two methyl groups are attached to carbon 3 (3,3-dimethyl).
Thus, the correct name is 3,3-dimethylpent-4-enal.

<answer>
\boxed{3,3-dimethylpent-4-enal}
</answer>"
```

**HSR diagnosis:** The failure is *comprehension* of IUPAC chain selection rules (parent-chain identification when both a C=O and an alkene must be included), not mechanism recall. HSR improved mechanistic clarity but did not apply a naming-level repair to flip from `pent-*` to the `hex-*` parent required by the Gold answer.

---

### QAIR (Quality-Aware Iterative Review)

**Checks recorded (as reflected in the run text):**

```
- Mechanistic plausibility (elimination -> vinyl ether ->
    Claisen): PASSED
- Role of NaHCO3 as neutralizing base (sulfenic acid): PASSED
- Internal consistency of proposed names vs. drawn skeleton:
    PASSED
- IUPAC audit: parent-chain selection and lowest-locant C=C
    (REQUIRED): SKIPPED
```

#### QAIR Error Note

**Missed mandatory audit.** QAIR should have enforced:

- Parent-chain audit (aldehyde rule): chain includes C=O and is maximized;

- Lowest-locant audit (C=C) within that parent chain;

- $[3, 3]$ - mapping check to determine which branch becomes part of the main chain.

Skipping these allowed a self-consistent but **wrong** `pent-*` narrative to pass.

**Observed plateau:** QAIR converged on a self-consistent *pent*-chain narrative and terminated without running the naming audit that would force re-evaluation of the parent chain under aldehyde rules.

*Conclusion kept by QAIR:*

```
"... The major product is 4,4-dimethylpent-5-enal ...
<answer> \boxed{4,4-dimethylpent-5-enal} </answer>"
```

## A.7 HSR/QAIR SUCCESS CASE ANALYSIS

This section highlights that HSR and QAIR can independently correct reasoning errors even when Monitor-RAG is not triggered. Each subsection provides a real HLE question and shows how HSR and QAIR independently correct the Baseline solvers errors without retrieval.

We present five representative success cases covering the main error types that HSR and QAIR are designed to repair: (i) missing or inconsistent reasoning steps, (ii) arithmetic or numerical inconsistencies, (iii) methodological misapplication, and (iv) insufficient peer-evidence consolidation.

### A.7.1 TYPE 1: REPAIRING MISSING OR INCONSISTENT REASONING STEPS

**Case 1: Pond predator reasoning**

**HLE Question:** All ponds are cement boxes (2 ft deep; 10 ft or 30 ft square). Half were built one year ago and half five years ago. Older ponds now host established insect communities. With a single emergence trap, choose the pond posing the greatest medical threat.

**Answer Choices (A–E):**

- A. 10 ft square, one year old.

- B. There will be no difference between ponds.

- **C. 30 ft square, one year old.**

- D. 10 ft square, five years old.

- E. 30 ft square, five years old.

---

**Baseline Trajectory (Incorrect)**

**Baseline summary.** The Baseline solver focuses on temperature rather than predators:

```
"Older ponds have more insects, so they host more mosquitoes.
Cooling is slower in small ponds; hence the 10-ft, 1-year pond
poses the greatest threat."
```

**Baseline Error Note**

**Missing predator chain.** The solution ignores the explicit cue that older ponds already host insect communities (including predators), so young ponds are more dangerous.
**Irrelevant heuristic.** It applies a small pond warms faster temperature heuristic that is not meaningful for uniform 2-ft cement boxes.

---

**HSR (Hierarchical Solution Refinement)**

**Reconstructed causal chain.** HSR supplies the missing ecological steps:
- Young ponds lack established predators; older ponds have predator communities.
- Larger surface area provides more larval habitat (30 ft: $30^2 = 900\,\text{ft}^2$, vs. $10^2 = 100\,\text{ft}^2$).
- Uniform depth and cement walls remove the small warms faster argument.

*HSR anchor:*

```
"Young = predator-free. Larger surface area -> more egg-laying
substrate. The 30-ft pond provides 9x the larval habitat, so the
30-ft, 1-year pond poses the greatest medical threat."
```

---

**QAIR (Quality-Aware Iterative Review)**

**Checks logged as performed:**

```
Performed:
- Predator-control logic (young vs old ponds) ........ PASS
- Surface-area calculation (900 vs 100) ............. PASS
- Temperature/structure audit for cement tanks ....... PASS
```

**Outcome.** HSR+QAIR repair the missing predator/surface-area reasoning and flip the answer from the incorrect 10-ft pond to the correct 30-ft, 1-year pond (C).

```
Final: "... choose the young 30-ft pond; predators colonize older
ponds, and 9x surface area increases larval habitat ..."
<answer>\boxed{C}</answer>
```

---

**Case 2: Predator-vision framing**

**HLE Question:** Twenty untrained undergraduates rank visual similarity of *Bombus* species from field photos; the researcher clusters mimicry syndromes based on that consensus. Is this approach valid for the ecological function of mimicry?

---

**Baseline Trajectory (Incorrect)**

**Baseline summary.**

```
"Human perception is close to predator perception, so the method
is acceptable for evaluating mimicry."
```

**Baseline Error Note**

**Missing ecological reasoning.** The solver assumes human and predator perception are interchangeable, ignoring UV/tetrachromatic vision in predators and the role of learned avoidance. It also neglects motion and background cues important for aposematic displays.

---

**HSR (Hierarchical Solution Refinement)**

**Repaired reasoning.** HSR inserts the missing ecological context:

- Many predators see UV or have different colour channels than humans.
- Predator avoidance is learned; untrained students lack aversive conditioning.
- Static photographs omit motion and environmental background.

*HSR anchor:*

```
"Undergraduate rankings can be a useful starting point for
hypotheses, but ecological validity requires predator-vision
models, multispectral imaging, or behavioural assays. Thus the
method is only partially valid."
```

---

**QAIR (Quality-Aware Iterative Review)**

**Checks logged as performed:**

```
Performed:
- Partial-utility acknowledgement (hypothesis generation) ....
    PASS
- Need for predator-vision / behavioural evidence ...........
    PASS
```

**Outcome.** HSR+QAIR convert an overconfident fully valid judgment into a qualified partially valid conclusion, correctly reflecting the ecological limitations of human-based similarity rankings.

```
Final: "Human rankings generate hypotheses; ecological validity
still requires predator-vision tests."
<answer>Partially valid</answer>
```

---

### A.7.2 TYPE 2: RESOLVING ARITHMETIC AND NUMERICAL INCONSISTENCIES

**Case 3: Ozonolysis atom tracking**

**HLE Question:** Follow cis-2-butene-1,4-diol (11) through TES protection, ozonolysis, $MeNO_2$ addition (nitroolefin 10), HayashiJrgensen addition (7), Nef oxidation (14), and final ring formation to multifidene (1). How many of 11s carbons reach multifidene (1)? How many oxygens from 11 are in 14? How many nitrogens from 7 appear in 10?

---

**Baseline Trajectory (Incorrect)**

**Baseline summary.** The Baseline solver implicitly treats compound 11 as if both TES-bearing fragments and nitrogen from 7 survive into later intermediates, leading to an overcount of carbons, oxygens, and N-origin.

> **Baseline Error Note**
>
> **Atom-count inconsistency.** Ozonolysis is treated as topology-preserving rather than fragmenting 11 into two $C_2$ pieces, and nitrogen in 10 is incorrectly traced back to 7 instead of $MeNO_2$.

---

**HSR (Hierarchical Solution Refinement)**

**Fragment-by-fragment repair.**

- Ozonolysis splits 11 into two $C_2$ fragments; only one TES-bearing fragment is carried forward 11 $\to$ 12 $\to$ 10 $\to$ 13 $\to$ 14 $\to$ 1.
- Nitroolefin 10 is formed *before* adduct 7; its nitrogen comes from $MeNO_2$, not 7.
- Each molecule of 14 therefore inherits two carbons and one oxygen from 11, and zero nitrogens from 7.

*HSR anchor:*

```
"After O3 cleavage, only one TES-bearing fragment enters the
sequence 12 -> 10 -> 13 -> 14 -> 1. Nitrogen in 10 is introduced
by MeNO2, not adduct 7. Thus each 14 carries 2 carbons and 1
    oxygen
from 11, and 0 nitrogens from 7."
```

---

**QAIR (Quality-Aware Iterative Review)**

**Checks logged as performed:**

```
Performed:
- Fragment sketch and mapping 11 -> 14 ............ PASS
- Source of nitrogen in 10 (MeNO2 vs 7) .......... PASS
- Final tally verification ....................... PASS
```

**Outcome.** HSR+QAIR resolve the numerical inconsistency and yield the correct atom-count triple (2, 1, 0).

```
Final: "Correct tally: 2 carbons, 1 oxygen, 0 nitrogens."
<answer>2, 1, 0</answer>
```

### A.7.3 TYPE 3: CORRECTING METHODOLOGICAL ERRORS

**Case 4: Jensen-guided allocation**

**HLE Question:** Given a strictly concave or convex benefit function $s(r_i)$ and total resources $R$, several statements relate to when a fair (equal split) vs. extreme (all-or-nothing) allocation is optimal. Which option lists all true statements?

**Answer Choices (A–P):**

- A. [1]

- B. [2]

- C. [3]

- **D. [4]**

- E. [1, 2]

- F. [1, 3]

- G. [1, 4]

- H. [2, 3]

- I. [2, 4]

- J. [3, 4]

- K. [1, 2, 3]

- L. [1, 2, 4]

- M. [1, 3, 4]

- N. [2, 3, 4]

- O. [1, 2, 3, 4]

- P. [5]

---

**Baseline Trajectory (Incorrect)**

**Baseline summary.**

```
"If s is strictly increasing, the fair allocation is optimal.
Therefore multiple statements about fair splitting are accepted
together."
```

**Baseline Error Note**

**Method choice error.** The solver conflates monotonicity with curvature, using informal heuristics instead of a principled method such as Jensens inequality plus counterexamples.

---

**HSR (Hierarchical Solution Refinement)**

**Methodological repair.** HSR replaces heuristics with a standard functional-analytic argument:
- For concave $s$, Jensens inequality implies that the equal split maximizes $\sum_i s(r_i)$ given $\sum_i r_i = R$.
- For convex $s$, extreme allocations can be optimal.
- Explicit counterexamples are constructed to show why statements 13 are not necessarily true.

*HSR anchor:*

```
"Apply Jensen: for concave s, sum s(r_i) <= n * s(R/n), with
equality at the fair split. For convex s, extremes dominate.
We can build explicit counterexamples for Statements 13,
leaving only Statement 4 true."
```

---

**QAIR (Quality-Aware Iterative Review)**

**Checks logged as performed:**

```
Performed:
- Jensen inequality citation and use ............ PASS
- Counterexample construction for 13 ........... PASS
- Mapping surviving statement -> option D ....... PASS

Final: "Only Statement 4 survives; therefore option D."
<answer>\boxed{D}</answer>
```

**Outcome.** HSR+QAIR correct the methodological choice, moving from heuristic monotonicity reasoning to a Jensen-based analysis that selects the correct option D.

```
Final: "Only Statement 4 survives; therefore option D."
<answer>\boxed{D}</answer>
```

---

### A.7.4 TYPE 4: PEER-EVIDENCE CONSOLIDATION AND STABILIZATION

**Case 5: Camera-trap augmentation**

**HLE Question:** A camera trap monitors a 7 m ravine using a 50 mm lens at $f/4$, ISO 190, and $1/500$ s with fixed infrared (IR) lighting. Training data come from colourful GBIF photos. Which augmentation is most important for deployment accuracy?

---

**Baseline Trajectory (Incorrect)**

**Baseline summary.** The Baseline solver focuses on optical artifacts alone and proposes blur-based augmentations (Gaussian or motion blur), without addressing the RGB $\rightarrow$ IR domain shift between GBIF training images and monochrome deployment images.

**Baseline Error Note**

**Peer evidence under-used.** Peer arguments about depth of field and motion blur are considered in isolation, and the much larger spectral mismatch (colour vs. infrared) is not treated as the dominant shift.

---

**HSR (Hierarchical Solution Refinement)**

**Peer Evidence Integration**

- *Peer A (Gaussian blur).* Provides depth-of-field estimates ($\approx 3.04.2$ m sharp zone at $f/4$) and argues for Gaussian blur to mimic defocus.
- *Peer B (motion blur).* Argues that animal motion can still cause blur at $1/500$ s and advocates motion-blur augmentation.

HSR re-evaluates these contributions and contrasts them with the RGB→IR domain gap:
- Training images: colourful RGB web photos.
- Deployment images: IR-illuminated, often monochrome or strongly colour-suppressed.
- Optical blur differences are smaller than spectral differences.

*HSR anchor:*

```
"Infrared camera-trap images are colour-deficient, whereas GBIF
training images are colourful RGB. Augmentations that disrupt
    colour
patterns, such as colour noise (option J), force the model to
    rely on
shape/texture rather than hue and directly address the dominant
RGB->IR domain shift."
```

---

**QAIR (Quality-Aware Iterative Review)**

**Checks logged as performed:**

```
Performed:
- DOF and motion-blur calculations acknowledged .... PASS
- RGB->IR domain-shift identification ............. PASS
- Justification of colour-noise as dominant fix .... PASS
```

**Outcome.** HSR+QAIR consolidate peer evidence, keep the blur calculations explicit, but ultimately choose colour-noise augmentation (J) as the most important transformation for bridging the training-to-deployment domain gap.

```
Final: "Colour noise (J) corrects the major RGB->IR discrepancy."
<answer>\boxed{J}</answer>
```

---

## A.8 REASONING-PROCESS REPAIR CASE ANALYSIS

For reasoning-process error, HSR (Hierarchical Solution Refinement) and QAIR (Quality-Aware Iterative Review) jointly provide the full correction mechanism. They repair reasoning defects by (i) reconstructing incomplete or partially incorrect causal chains, (ii) correcting invalid logical inferences

even when all relevant facts are already present, and (iii) detecting and resolving subtle inconsistencies introduced during refinement.

### A.8.1 CASE 1: HSR CHAIN RECONSTRUCTION

**HLE Question:** An elderly patient treated for insomnia and seizures was switched among several antiepileptic drugs. After the third drug was started, the patient developed severe skin blistering. Which specific biochemical reaction initiated the cascade leading to this blistering reaction?

---

**Baseline Trajectory (Incorrect)**

**Baseline summary.**

```
"The development of skin blisters is consistent with a SCAR such
    as SJS.
The specific biochemical reaction ... involves metabolism of
    lamotrigine
by cytochrome P450 enzymes (particularly CYP3A4) to form a
    reactive
arene oxide on its dichlorophenyl ring."
```

**Baseline Error Note**

**Incomplete causal chain.** The baseline names lamotrigine oxidation, but omits (i) valproate-mediated UGT inhibition, (ii) the forced diversion into a minor P450 route, and (iii) the downstream immune cascade responsible for SJS/TEN lesions.

---

**HSR (Hierarchical Solution Refinement)**

**Reconstructed mechanism.** HSR completes the causal chain:

- **UGT blockade.** Coadministration of valproic acid inhibits lamotrigine glucuronidation (UGT1A4/UGT2B7), removing its main detox pathway.

- **Forced oxidation.** Lamotrigine is diverted toward cytochrome P450-mediated bioactivation (primarily CYP2A6/CYP2D6) to a reactive arene oxide.

- **Immune trigger.** When epoxide hydrolase and glutathione detox pathways are saturated, arene oxide accumulates, haptenizes keratinocyte proteins, and initiates the SJS/TEN immune cascade.

*HSR anchor:*

```
"Valproate shuts down UGT glucuronidation, so lamotrigine is
    oxidised
by CYP2A6/2D6 to a reactive arene-oxide. When epoxide
    hydrolase/GSH
are saturated, the arene-oxide binds keratinocyte proteins,
    forming
the hapten that initiates the SJS/TEN immune cascade."
```

---

**QAIR (Quality-Aware Iterative Review)**

**Checks logged as performed:**

```
Performed:
- Mechanism chain completeness ................. PASS
- Clinical-sequence summary .................... PASS
Requested clarifications:
- Acknowledge CYP2A6/2D6 as key minor-pathway enzymes
- Trim tangential discussion; focus on initiating oxidation
```

**Outcome.** After HSR incorporates the enzyme clarifications and removes tangential details, QAIR accepts the refined causal narrative as complete and mechanistically consistent.

```
Final:
<answer>
Cytochrome P450mediated oxidation of lamotrigine
(after valproate blocks UGT1A4/UGT2B7) to a reactive
arene oxide that overwhelms detox pathways and
triggers SJS/TEN blistering.
</answer>
```

### A.8.2 CASE 2: HSR INFERENCE CORRECTION

**HLE Question:** A hypothetical population is infinitely large, randomly mating, diploid, and experiences no mutation, migration, or selection; all genotypes have equal fitness and reproduce in discrete generations. A quantitative trait on an autosomal chromosome has broad-sense heritability $H^2 = 0.7$, and a researcher measures all individuals in the western and eastern halves of the region at a single time point. Which statements must always be true?

**Answer Choices (A–P):**

- A. 1 only

- B. 2 only

- C. 3 only

- D. 4 only

- E. 1 and 2 only

- F. 1 and 3 only

- G. 1 and 4 only

- H. 2 and 3 only

- I. 2 and 4 only

- J. 3 and 4 only

- K. 2, 3, and 4 only

- L. 1, 3, and 4 only

- M. 1, 2, and 4 only

- N. 1, 2, and 3 only

- O. 1, 2, 3, and 4

- P. None of the options must always be true

---

**Baseline Trajectory (Incorrect)**

**Baseline summary.**

```
"With random mating, no spatial structure, and an infinite
    population,
the genetic composition is uniform, leading to no substantial
    geographic
variation in phenotype. ... Thus, statements 1, 3, and 4 must
    always be
true, so choose option [1,3,4]."
```

**Baseline Error Note**

**Facts correct, inference wrong.** The baseline equates uniform genetics with uniform phenotype, ignoring that $H^2 = 0.7$ implies 30differ even in an infinite population.

---

---

**HSR (Hierarchical Solution Refinement)**

**Repaired reasoning.** HSR separates givens from conclusions:

- No selection, mutation, or gene flow $\Rightarrow$ statement 1 is necessarily true (no selection on the measured trait).

- Infinite size and no mutation/gene flow remove drift and differentiation, but $H^2 < 1$ leaves 30% environmental variance, so west–east mean phenotypes need not be identical; statement 4 is not forced.

- With no mutation and no gene flow, the conditions for future speciation are absent, so statement 3 is necessarily true.

*HSR anchor:*

```
"Even in an infinite HW population, H^2 = 0.7 still leaves 30%
environmental variance, so West vs East means can differ.
    Therefore
only statements 1 and 3 are forced; select option F."
```

---

**QAIR (Quality-Aware Iterative Review)**

**Checks logged as performed:**

```
Performed:
- Reasoning-rule validity ....................... PASS
- Weighting sanity check (genetic vs env) ........ PASS
- Multi-evidence reconciliation .................. PASS
Notes:
- Make explicit that no mutation/gene flow prevents speciation
- Highlight that H^2=0.7 => 30% environmental variance
```

**Outcome.** QAIR confirms that the corrected solution uses heritability and population-genetic assumptions appropriately, and that only statements 1 and 3 must hold.

```
Final: "Only statements 1 and 3 are forced; therefore option F."
<answer>\boxed{F}</answer>
```

---

### A.8.3 CASE 3: QAIR CONSISTENCY DETECTION

**HLE Question:** Five alien species differ in external traits such as setae, color, claws, simple eyes, and antenna morphology. Using these characters, construct a trait-difference matrix and infer the most parsimonious phylogeny. Which tree (in numeral notation, grouping sister taxa with parentheses) is the most parsimonious?

**Answer Choices (A–E):**

- A. (((1,(4,2)),3),5)

- B. (3,(2,(4,(1,5))))

- C. ((3,((4,1),2)),5)

- D. ((1,4),5) (2,3)

- E. (((1,4),(2,3)),5)

---

**Baseline Trajectory (Incorrect)**

**Baseline summary.**

```
"Tree A ... 7 changes ... Tree C ... 6 changes ... Tree E ... 7
    changes.
Tree C has the lowest parsimony score of 6, making it the most
parsimonious phylogeny."
```

**Baseline Error Note**

**Unsupported conclusion.** The baseline asserts that Tree C has 6 steps and is most parsimonious, but does not show the underlying trait matrix or per-tree Fitch step counts, so the result is unverifiable.

## HSR (Hierarchical Solution Refinement)

**First-pass refinement (still incomplete).** HSR attempts to fill in missing structure:
- Introduces a binary trait matrix for the five morphological characters (texture, simple eye, antennae, serration, claws).
- Claims updated parsimony scores, still reporting Tree C with 6 changes and alternatives with higher scores.

**Residual HSR Error**

**Still incomplete.** The initial HSR pass mis-codes certain character states (e.g., species 2 antennae) and fails to provide explicit Fitch step-by-step counts, leaving the parsimony claim unverifiable.

## QAIR (Quality-Aware Iterative Review)

**QAIR detection and enforced repair.**

```
Performed:
- Consistency audit ............................. FAIL
- Character-coding check ........................ FAIL
- Per-tree step count audit ..................... FAIL
Action required:
- Provide a corrected trait matrix
- Add explicit Fitch step counts for every topology
```

**QAIR Error Note**

**Unverifiable reasoning.** QAIR determines that the refinement still lacks a correct trait matrix and per-tree Fitch counts, meaning the claim that "Tree C has 6 changes" is not yet logically or empirically supported. QAIR therefore blocks finalization and demands explicit reconstruction.

Following QAIRs requirements, HSR revises the solution:
- Cleans the trait table for all five characters across taxa.
- Runs explicit Fitch parsimony evaluation, showing: Tree C has 6 changes; Trees A and E have 7; others $> 7$.
- Attaches the per-tree step count so that the "6 vs. 7" claim is auditable.

*QAIR-approved summary:*

```
"Trait table cleaned (texture, simple eye, antennae, serration,
    claws).
Fitch evaluation shows Tree C = 6 changes; Trees A and E = 7;
    others > 7.
Therefore Tree C is the unique most parsimonious topology."
```

**Outcome.** Only after the corrected matrix and full Fitch counts are provided does QAIR allow the solution to finalize.

```
Final: "Tree C is the unique most parsimonious tree with 6
    changes."
<answer>\boxed{C}</answer>
```

## A.9 COMPONENT-ERROR MAPPING

Table A3: Mapping between error types and repairing modules

| Error Type | Monitor-RAG | HSR | QAIR |
|---|---|---|---|
| Knowledge insufficiency | **Primary fix** | — | — |
| Reasoning-chain errors | Flags uncertainty | **Primary fix** | Secondary consistency check |
| Numerical / arithmetic errors | — | Targeted correction | Re-verification |
| Methodological errors | Sometimes helpful | **Strong fix** | Filter for incorrect revisions |
| Overconfidence / incomplete justification | — | — | **Final quality check** |

## A.10 ADDITIONAL BENCHMARK RESULTS

Table A4: Benchmark comparison on HLE Bio/Chem (149 problems; `o3-mini` judge)

| Model | Acc (%) |
|---|---|
| *LLMs* | |
| DeepSeek V3.1 (Non-Think) | 6.71 |
| Deekseek R1 | 10.74 |
| Qwen3 235B A22B | 15.38 |
| *LLM with Tools* | |
| Deekseek R1 with Browsing | 16.82 |
| DeepSeek V3.1 with Browsing | 11.21 |
| Doubao with Browsing | 11.21 |
| *Agents* | |
| Kimi Researcher | 9.35 |

## A.11 ADDITIONAL EXPERIMENTS WITH A GENERIC MIXED-DOMAIN RETRIEVAL CORPUS

To examine whether the improvements of Eigen-Agent primarily arise from its reasoning mechanisms rather than domain alignment between the retrieval corpus and the evaluation benchmarks, we performed additional experiments using a generic mixed-domain retrieval corpus, along with a configuration that disables retrieval entirely.

The generic retrieval corpus was constructed by sampling 14,731 post-2020 papers from the Hugging-Face `arxiv-papers` dataset[2], which spans 148 subject areas. For each paper, we extracted concise knowledge snippets from the abstract using a unified prompt template (shown below). The resulting corpus is intentionally heterogeneous and contains no explicit alignment with the biology or chemistry domains of the evaluation benchmarks.

> **Prompt of Generic Corpus Construction**
>
> You are an expert assistant helping extract high-level claims from scientific paper abstracts.
> Task: Given a scientific abstract, extract ONLY the statements, insights, or claims explicitly present in the abstract itself. Do NOT invent, guess, or add any extra content that is not written in the abstract. Only include knowledge points that are clearly stated.
> **Do NOT include:**
> - The method proposed by the paper (e.g., We propose, Our model)

---

[2]https://huggingface.co/datasets/nick007x/arxiv-papers

- Implementation details or experimental setup
- Numerical results or performance comparisons
- Dataset descriptions
- Any statements uniquely tied to this specific papers approach, model, or results
- Anything not explicitly stated in the abstract

**DO include:**

- Background claims
- Scientific facts and relationships
- Observations, hypotheses, or general principles explicitly mentioned
- Statements about phenomena, mechanisms, trends, limitations, or open problems explicitly mentioned

**Additional requirements:**

- Do NOT repeat the same claim. Make each claim distinct.
- If multiple sentences convey the same insight, combine them into a single concise claim.
- If there are NO claims in the abstract, output exactly: { "claims": [] }

Important formatting rule: Always output JSON only, with this exact shape:

```
{
    "claims":
        [
            "claim 1",
            "claim 2",
            ...
        ]
}
```
Do NOT output any extra explanation or text outside the JSON object.

We evaluated Eigen-Agent under two new configurations: a no-RAG setting where only HSR and QAIR operate, and retrieval over the generic mixed-domain corpus. As shown in Table A5, Eigen-Agent retains strong performance even when retrieval is entirely disabled, indicating that the iterative correction mechanisms in HSR and the quality gating in QAIR contribute significantly to overall accuracy. When supplied with a generic mixed-domain corpus, the system achieves substantial improvements, suggesting that Eigen-Agents mechanisms are robust to corpus heterogeneity and noise. Retrieval over the Bio/Chem corpus provides an additional but modest improvement, which is consistent with the expected benefits of specialized background knowledge. Together, these findings confirm that the performance gains of Eigen-Agent are driven primarily by its architectural design and not by privileged domain-specific retrieval.

Table A5: Evaluation of Eigen-Agent under no-retrieval, generic mixed-domain, and Bio/Chem retrieval corpora.

| Configuration | Acc (%) |
| --- | --- |
| Baseline | 25.3% |
| **Baseline + HSR/QAIR (No RAG)** | **34.5%** |
| **Eigen-Agent (Generic Mixed-Domain Corpus)** | **45.6%** |
| Eigen-Agent (Bio/Chem Domain Corpus) | 48.3% |

## A.12 PSEUDO-CODE

---

**Algorithm 1** Eigen-Agent: High-level Workflow

---

**Require:** Query $q$, config $\mathcal{C}$ (LLM & retriever), #proposers $K$ (e.g., 5), QAIR threshold $\tau$, max rounds $T_{\max}$
**Ensure:** Final solution $s^\star$
1: **Init:** set up LLM, RETRIEVER, and tool endpoints from $\mathcal{C}$
2:
    **Proposer generates initial solutions**
3: **for all** $i \in \{1, \ldots, K\}$ **in parallel do**
4:     $S[i] \leftarrow \textsc{Generate}(q)$               ▷ initial solution generation
5: **end for**
6:
    **Local correction (optional, per-candidate)**
7: **for all** $i \in \{1, \ldots, K\}$ **in parallel do**
8:     $C[i] \leftarrow \textsc{Corrector}(S[i])$         ▷ targeted fixes without cross-solution access
9: **end for**
10:
    **Hierarchical Solution Refinement (HSR)**
11: $R \leftarrow \varnothing$
12: **for** $a \in \{1, \ldots, K\}$ **do**              ▷ rotate anchors
13:     $A \leftarrow C[a], \quad Ref \leftarrow C \setminus \{C[a]\}$
14:     $R[a] \leftarrow \textsc{Refine}(A, Ref)$    ▷ apply peer-informed repairs: logic, numeric, method, expression
15: **end for**
16:
    **Quality-Aware Iterative Reasoning (QAIR)**
17: $t \leftarrow 0, \quad P \leftarrow R$
18: **while** $t < T_{\max}$ **do**
19:     **(parallel)** for each $s \in P$: $(q_{\text{logic}}, q_{\text{ans}}, q_{\text{exp}}, suggestion) \leftarrow \textsc{Evaluator}(s)$
20:     $score(s) \leftarrow 0.2 \cdot q_{\text{logic}} + 0.6 \cdot q_{\text{ans}} + 0.2 \cdot q_{\text{exp}}$
21:     $Pass \leftarrow \{s \in P \mid score(s) \geq \tau\}; \quad Fail \leftarrow P \setminus Pass$
22:     **if** $Fail = \varnothing$ **then break**
23:     **end if**
24:     **(parallel)** for each $s \in Fail$: $\tilde{s} \leftarrow \textsc{Corrector}(s, suggestion)$
25:     $P \leftarrow Pass \cup \{\tilde{s} \mid s \in Fail\}; \quad t \leftarrow t + 1$
26: **end while**
27:
    **Rank & select**
28: $s^\star \leftarrow \textsc{Ranker.Select}(P)$         ▷ e.g., composite score or pairwise compare
29: **return** $s^\star$

    **Subroutines used in all LLM generation process**
30: **function** MONITOR-BASED RAG$(q)$
31:     $ctx \leftarrow \textsc{InitContext}(q)$
32:     **while** not DONE$(ctx)$ **do**
33:         $ctx \leftarrow \textsc{LLM.Next}(ctx)$
34:         **if** MONITOR$(ctx) = 1$ **then**     ▷ detect uncertainty/insufficiency on-stream
35:             $qry \leftarrow \textsc{Querier}(ctx)$     ▷ minimal, targeted keywords
36:             $docs \leftarrow \textsc{Retriever}(qry)$
37:             $ctx \leftarrow \textsc{Injector}(ctx, docs)$     ▷ compress & integrate evidence seamlessly
38:         **end if**
39:     **end while**
40:     **return** TRACETOSOLUTION$(ctx)$
41: **end function**

---

## A.13 USAGE OF LANGUAGE MODELS

We utilized a large language model (LLM) to aid in the preparation of this manuscript. Its use was limited to editorial tasks, including proofreading for typographical errors, correcting grammar, and improving the clarity and readability of the text.

