# OpenReview forum: "Eigen-Agent: Adaptive Multi-Agent Scientific Reasoning with Monitor-Based RAG"
_ICLR.cc/2026/Conference — ICLR 2026 Poster_

### Official Review · Reviewer_nD2g · 2025-10-30

**Soundness:** 3
**Presentation:** 3
**Contribution:** 3
**Rating:** 6
**Confidence:** 4

**Summary:**

This paper presents EIGEN-1, a multi-agent framework for scientific reasoning that integrates three modules:
(1) Monitor-based RAG, an implicit retrieval mechanism that adaptively augments reasoning without explicit tool calls;
(2) Hierarchical Solution Refinement (HSR), which coordinates agents through an anchor–reference structure for iterative correction; and
(3) Quality-Aware Iterative Reasoning (QAIR), a quality-gated refinement loop.
The framework is evaluated on HLE Bio/Chem, SuperGPQA, and TRQA datasets, showing clear gains over strong LLM baselines.
The paper is clearly written, well-structured, and the empirical results are strong.

**Strengths:**

1.Comprehensive System Design:
The integration of retrieval, multi-agent collaboration, and quality-aware refinement forms a coherent and practically useful system.The proposed orchestration shows engineering sophistication and appears well implemented.

2.Strong Empirical Results:
Across multiple benchmarks, EIGEN-1 demonstrates consistent improvements in accuracy and efficiency.The ablation analyses (e.g., removing Monitor or QAIR) are detailed and convincing.

3.Deployment Practicality:
Since the framework operates over frozen LLMs without training, it can be directly applied to existing models, which increases its practical relevance.

**Weaknesses:**

Domain-specific RAG corpus and evaluation validity

The paper’s most significant issue lies in how the retrieval corpus was constructed and evaluated.The authors build a biology/chemistry-specific RAG database from domain papers (Appendix A.1) and test primarily on HLE Bio/Chem and SuperGPQA Bio datasets.
However, the HLE benchmark itself explicitly states that:
“Each question has a known solution that is unambiguous and easily verifiable, but cannot be quickly answered via internet retrieval.”

This creates a conceptual inconsistency: the proposed system demonstrates large gains precisely on a dataset that was designed to minimize the benefits of retrieval. Even if the database does not contain direct answers, its domain overlap likely gives the model privileged access to related context or background information. Thus, the reported improvements may reflect content alignment rather than genuine reasoning enhancement.

I strongly recommend the authors to: Report results using a generic, mixed-domain corpus (e.g., Wikipedia or full arXiv); Or evaluate cross-domain performance, e.g., using the Bio/Chem corpus for non-scientific HLE subsets. These analyses would clarify whether EIGEN-1’s gains stem from its mechanism or its domain priors.

**Questions:**

Refer to the weaknesses.

---

> ### Author Response · Authors · 2025-11-20
>
> Dear Reviewer,
>
> Thank you sincerely for your thoughtful and rigorous assessment of our work. We are grateful for your positive remarks on the overall design and empirical strength of Eigen-1, and we especially appreciate your careful scrutiny of the retrieval corpus and evaluation validity. Your comments prompted us to conduct substantial additional analyses, which we summarize below.
>
> ---
>
> # 1. Response to Concern: Validity of Using a Domain-Specific RAG Corpus on HLE & Risk of Content Alignment
>
> **Your concern:**
> > *“The paper’s most significant issue lies in how the retrieval corpus was constructed and evaluated.
> > The HLE benchmark explicitly states that: ‘Each question … cannot be quickly answered via internet retrieval.’
> > Yet the system shows large gains with a domain-specific Bio/Chem corpus.
> > Even if the database does not contain direct answers, its domain overlap likely gives the model privileged access or content alignment.”*
>
> We fully appreciate this concern and agree that the distinction between **background knowledge** and **answer leakage** must be clarified. Our responses and additional analyses are as follows.
>
> ### (a) What the HLE restriction targets
> The HLE statement is meant to prevent *direct lookup* of question–answer pairs (e.g., solutions on web Q&A sites).
> It does **not** forbid retrieving domain background knowledge such as:
>
> - chemical reaction principles,
> - genetic drift definitions,
> - ecological mechanisms, etc.
>
> For graduate-level scientific reasoning tasks, such knowledge is often indispensable.
>
> ### (b) Manual audit: **0 cases of answer leakage**
> We conducted a line-by-line human audit of all retrieved passages across HLE Bio/Chem tasks.
>
> - **0 passages** contained the final answer, an answer choice, or a matching derivation.
> - Retrieved content consisted exclusively of general background knowledge (reaction patterns, formula definitions, physiological concepts, etc.).
>
> The model still performs multi-step reasoning, combining these pieces to reach the correct answer.
>
> ### (c) Full transparency
> To ensure community verification and eliminate any ambiguity,
> **we will open-source the entire RAG corpus and retrieval index** at camera-ready.
>
> **Conclusion:**
> The gains reflect improved reasoning supported by background knowledge, not memorization or content leakage.
>
> ---
>
> # 2. Response to Concern: Need for Mixed-Domain / Cross-Domain Retrieval Evaluation
>
> **Your concern:**
> > *“I strongly recommend the authors to:
> > (1) report results using a generic, mixed-domain corpus (e.g., Wikipedia, full arXiv), or
> > (2) evaluate cross-domain performance.
> > These analyses would clarify whether Eigen-1’s gains stem from its mechanism or its domain priors.”*
>
> We followed your recommendation and conducted substantial new experiments.
>
> ---
>
> ## 2.1 New Experiment: **Generic Mixed-Domain RAG Corpus (No Bio/Chem Alignment)**
>
> We constructed a *Generic RAG Corpus* by sampling **14,731 post-2020 papers** from the HuggingFace `arxiv-papers` dataset, covering **148 distinct subject areas**.
> Knowledge snippets were extracted from abstracts using GPT-4o.
>
> We then evaluated EIGEN-1 under three retrieval conditions:
>
> | Configuration | Accuracy (%) | Interpretation |
> |---------------|--------------|----------------|
> | Baseline | 25.3% | No retrieval, no refinement |
> | **[New] Baseline + HSR/QAIR (No RAG)** | **34.5%** | Reasoning modules alone give large gains |
> | **[New] Eigen-1 (Generic Mixed-Domain RAG)** | **45.6%** | Mechanism still highly effective with no domain alignment |
> | Eigen-1 (Domain Bio/Chem RAG)** | 48.3% | Domain knowledge yields a modest additional boost |
>
> ### Key Findings
>
> #### (1) **Mechanism-driven improvements**
> Even **without any retrieval**, HSR+QAIR improve the baseline by **+9.2%**,
> and with **generic**, noisy retrieval, the system still improves by **+20.3%**.
>
> This shows that Eigen-1’s core innovations—Monitor-based implicit retrieval, HSR, QAIR—generate most of the performance gains.
>
> #### (2) **Domain knowledge adds only an incremental benefit**
> Using Bio/Chem RAG adds **+2.7%** over generic RAG.
> This is expected: specialized knowledge enhances scientific reasoning, but is not required for the mechanism to function effectively.
>
> #### (3) **No dependence on domain alignment**
> The mixed-domain corpus spans physics, CS, econ, math, and social sciences—
> far outside Bio/Chem, yet Eigen-1 still performs strongly.
> This confirms the **generalizability and robustness** of the architecture.
>
> ---
>
> # Final Conclusion
>
> Your suggestion significantly strengthened the paper.
> The new ablations demonstrate clearly that:
>
> - Eigen-1’s gains are **not** driven by content alignment or privileged domain overlap;
> - The architecture’s reasoning mechanisms are **dominant contributors** to performance;
> - Domain knowledge provides only a reasonable, modest improvement on top of an already robust system.

---

### Official Review · Reviewer_iFym · 2025-11-01

**Soundness:** 3
**Presentation:** 4
**Contribution:** 3
**Rating:** 6
**Confidence:** 3

**Summary:**

This paper presents EIGEN-1, a novel framework for scientific reasoning that unifies implicit retrieval with structured multi-agent refinement. The authors aim to address two major challenges in retrieval-augmented reasoning systems: the fragmentation of reasoning caused by explicit retrieval—referred to as the “tool tax”—and the dilution of strong candidate solutions through democratic multi-agent aggregation.

On the Humanity’s Last Exam (HLE) Bio/Chem Gold benchmark, EIGEN-1 achieves 48.3% accuracy, outperforming the strongest baseline model, SciMaster, by 13.4 points while reducing token usage by 53.5% and reasoning steps by 43.7%. Further evaluations on SuperGPQA and TRQA confirm its robustness. The paper also provides insightful analyses, including a quantitative study of the tool tax and the diversity–consensus trade-off in retrieval versus reasoning tasks.

**Strengths:**

1. The integration of implicit retrieval and hierarchical refinement is novel, bridging retrieval efficiency and collaborative reasoning in a unified system.

2. The modular design (Monitor-Querier-Injector-HSR-QAIR) is well-justified and empirically validated through controlled ablations and analyses.

3. The paper is clearly organized with intuitive visuals and detailed appendices (e.g., Algorithm 1).

4. The quantitative treatment of tool tax offers a new analytical lens for reasoning system efficiency, which can influence future research directions.

5. Strong benchmarks, token-efficiency metrics, and error-type breakdowns demonstrate robustness and depth of evaluation.

**Weaknesses:**

1. Lack of prompt sensitivity analysis.
Although the paper provides detailed templates for each component (Monitor, Querier, Injector, HSR, QAIR), it does not evaluate how sensitive the framework’s performance is to prompt wording, structure, or length. Since EIGEN-1 heavily relies on prompt-driven modular coordination, empirical evidence on prompt robustness would strengthen the paper’s claims of generality.

2. Limited exploration of base-model dependence.
All experiments are conducted on DeepSeek-V3.1 as the underlying model. The authors claim that Monitor-based RAG is “model-agnostic,” but they do not test the framework on other LLMs such as GPT-4.1 or Claude Opus. It remains unclear how performance, retrieval triggering frequency, or refinement behavior change when the base model’s reasoning or linguistic style differs.

3. Potential evaluator bias.
The QAIR and automatic judging components rely on LLM-based evaluators (e.g., o3-mini). Without cross-model validation, the risk of evaluator–solver coupling remains unquantified.

4. Domain focus.
The benchmarks are concentrated in biology and chemistry. Additional experiments in other reasoning domains (e.g., physics, math) would better establish the framework’s general applicability.

**Questions:**

1. How sensitive is EIGEN-1 to prompt variations? Have you observed significant degradation when the prompts for the Monitor, Querier, or QAIR modules are paraphrased or shortened?

2. Since the method is described as model-agnostic, could you report or estimate its performance when using a different base model (e.g., GPT-4, Claude 3, or Gemini 2.5)? Does the Monitor module’s uncertainty detection generalize across architectures?

3. QAIR and the o3-mini judge play central roles in assessment. How robust are the results if these evaluators are replaced by another LLM or a human scoring subset?

4. Do you expect similar gains in domains that require abstract reasoning but less factual grounding (e.g., logic puzzles, programming tasks)?

---

> ### Author Response · Authors · 2025-11-18
> **Response to Reviewer iFym (1/2)**
>
> Dear Reviewer,
>
> We thank you for your detailed review and valuable feedback on our work. We are especially grateful for your positive comments on the novelty of Eigen-1's integration of implicit retrieval and hierarchical refinement, the well-justified modular design, and the quantitative analysis of "tool tax."
>
> We have conducted several targeted supplementary experiments and analyses to address the four key points raised in your "Weaknesses" and "Questions." We are pleased to report that these new results substantially strengthen the claims made in our paper.
>
> &nbsp;
>
> ---
> &nbsp;
>
> #### 1. Response to Concern 1: Lack of prompt sensitivity analysis
>
> **Your Concern:**
> > You noted that Eigen-1 relies heavily on prompt-driven coordination, and a lack of robustness testing could mean the framework is "fragile" to prompt wording or structure.
>
> **Our Response:** We agree this is an important point. To verify the robustness of Eigen-1, we conducted sensitivity tests on the prompts for our two most critical components (Monitor and HSR). We created two variants:
> * **Paraphrased:** Used an LLM to rephrase the prompts, preserving semantics but altering syntax and wording.
> * **Concise:** Significantly shortened the prompts to retain only the core instructions.
>
> We ran these experiments on the full HLE benchmark. The results are as follows:
>
> | Prompt Version | Accuracy (%) | Change |
> | :--- | :--- | :--- |
> | Original | 48.3% | - |
> | **[New Experiment]** Paraphrased | 48.9% | +0.6% |
> | **[New Experiment]** Concise | 46.9% | -1.4% |
>
> **Conclusion:** The results show that Eigen-1 is **highly robust** to prompt variations. The paraphrased version even yielded a slight improvement (+0.6%), and the concise version resulted in only a minimal 1.4% dip. This demonstrates that our framework is not "fragile" and robustly handles reasonable changes in prompt design.
>
> &nbsp;
>
> ---
> &nbsp;
>
> #### 2. Response to Concern 2: Limited exploration of base-model dependence
>
> **Your Concern:**
> > You pointed out that all experiments were based on DeepSeek-V3.1, and our "model-agnostic" claim was not validated on other models (e.g., GPT-4.1 or Claude).
>
> **Our Response:** To validate the "model-agnostic" nature of Eigen-1 and the transferability of its architectural gains, we conducted a full set of supplementary experiments using a base model with a distinctly different architecture and style (o4-mini).
>
> We compared the Baseline (o4-mini) vs. Eigen-1 (o4-mini) performance and contrasted it with our original results on DeepSeek-V3.1:
>
> | Base Model | Configuration | Accuracy (%) | Relative Improvement |
> | :--- | :--- | :--- | :--- |
> | DeepSeek-V3.1 | Baseline | 25.3% | |
> | DeepSeek-V3.1 | **Eigen-1** | **48.3%** | **+23.0%** |
> | **[New Experiment]** o4-mini | Baseline | 20.1% | |
> | **[New Experiment]** o4-mini | **Eigen-1** | **41.6%** | **+21.5%** |
>
> **Conclusion:** The results are exceptionally clear. Although the o4-mini baseline is lower, the **relative performance improvement** gained from the Eigen-1 framework (+21.5%) is **remarkably consistent** with the gains observed on DeepSeek-V3.1 (+23.0%). This strongly supports our "model-agnostic" claim, proving that the architectural benefits (implicit RAG and HSR/QAIR) are transferable and not an artifact of a specific base model.
>
> &nbsp;
>
> ---
> &nbsp;
>
> #### 3. Response to Concern 3: Potential evaluator bias
>
> **Your Concern:**
> > You raised a critical point that both QAIR (internal evaluation) and the automatic judging (external evaluation) rely on o3-mini, risking an "evaluator-solver coupling".
>
> **Our Response:** To quantify and address this concern, we conducted two rigorous human-in-the-loop validation studies:
>
> **a) QAIR Internal Validation:**
> We sampled 25 representative problems (involving 30 QAIR decisions) and had 2 human experts evaluate the "pass/fail" decisions made by the QAIR module.
> * The human experts (`human_judge`) and the QAIR evaluator (`QAIR_judge`) disagreed on only 1 out of 30 decisions, achieving a **96.7% agreement rate**.
> * The **Spearman's rank correlation ($\rho$)** between the two sets of decisions is **0.935**.
>
> **b) Final Answer Automatic Judging Validation:**
> We randomly sampled 30 final answers and had 2 human experts judge their "correct/incorrect" status, comparing this to our automatic evaluator (`llm_judge`).
> * The human expert judgments (`human_judge`) were in **100% (30/30) agreement** with the `llm_judge`.
> * The **Spearman's rank correlation ($\rho$)** is **1.0**.
>
> **Conclusion:** These two studies demonstrate an exceptionally high correlation between our automated evaluators and human expert judgment. This strongly mitigates concerns about "evaluator-solver coupling" and confirms that o3-mini is a reliable proxy for human evaluation in this task.

---

> > ### Author Response · Authors · 2025-11-18
> > **Response to Reviewer iFym (2/2)**
> >
> > #### 4. Response to Concern 4: Domain focus
> >
> > **Your Concern:**
> > > You noted our benchmarks are concentrated in biology and chemistry, and asked if Eigen-1 would yield similar gains in domains requiring abstract reasoning with less factual grounding (e.g., logic puzzles, programming tasks).
> >
> > **Our Response:** This is a valuable question regarding generalization. We selected the biology and chemistry domains because the HLE Bio/Chem Gold benchmark represents a highly challenging class of problems in scientific reasoning. These problems are **extremely difficult** and are **typical examples of tasks that require Retrieval-Augmented Generation (RAG) to solve** — they demand a combination of deep domain knowledge (e.g., Watterson’s estimator) and complex multi-step reasoning.
> >
> > We believe that Eigen-1's significant success (+23.0% relative improvement) in this difficult and RAG-intensive domain robustly validates the effectiveness of our framework (implicit RAG and hierarchical refinement).
> >
> > Regarding the other domains you mentioned:
> > * For other scientific domains like **physics and mathematics**, we expect Eigen-1 to be highly applicable. Monitor-RAG can be used to implicitly retrieve formulas, theorems, or physical laws, while HSR/QAIR can correct logical errors in derivations.
> > * For domains with less factual grounding like **logic puzzles or programming tasks**, we agree that the Monitor-RAG component would be less active. However, as we have shown in our response to Reviewer 1, the **HSR and QAIR modules provide independent and robust logical correction capabilities even in the absence of RAG**.
> >
> > We appreciate this suggestion and will consider the migration of Eigen-1 to other scientific domains (such as physics or math) as important **future work**.
> >
> > We thank you again for your constructive feedback, which has significantly helped in improving the rigor and quality of our paper.

---

> ### Author Response · Authors · 2025-11-18
> **Response to Reviewer iFym (Questions)**
>
> For completeness, we also provide direct answers to the reviewer’s four questions:
>
> (1) **Prompt variation sensitivity:**
>
> >  it does not evaluate how sensitive the framework’s performance is to prompt wording, structure, or length
>
> As shown in our response on prompt robustness, both paraphrased (+0.6%) and concise (–1.4%) prompt variants yield minimal accuracy changes, indicating strong stability. It indicates that our performance improvement mainly comes from the framework design rather than the specially selected prompt.
>
> (2) **Performance on other base models:**
> > It remains unclear how performance, retrieval triggering frequency, or refinement behavior change when the base model’s reasoning or linguistic style differs.
>
>
> As reported in our cross-model experiment, o4-mini exhibits a similar relative improvement (+21.5%) to DeepSeek-V3.1 (+23.0%), supporting the framework’s model-agnostic behavior. This indicates that our framework design is not limited to a specific base model.
>
> (3) **Robustness of QAIR / o3-mini evaluation:**
>
> > Without cross-model validation, the risk of evaluator–solver coupling remains unquantified.
>
> Our human validation studies show 96.7% and 100% agreement with the automated evaluator, demonstrating no evaluator–solver coupling and confirming evaluator reliability. This demonstrates that the reliability of LLM Judge is guaranteed in specific use cases.
>
> (4) **Applicability to abstract reasoning domains:**
>
> > Additional experiments in other reasoning domains (e.g., physics, math) would better establish the framework’s general applicability.
>
>
> As discussed in our response on domain generalization, the architecture can naturally extend to physics/math (via implicit formula retrieval) and to logic/programming tasks where HSR+QAIR operates effectively even with minimal retrieval.

---

### Official Review · Reviewer_h3qW · 2025-11-01

**Soundness:** 3
**Presentation:** 3
**Contribution:** 2
**Rating:** 6
**Confidence:** 3

**Summary:**

This paper presents EIGEN-1, a new multi-agent system for scientific reasoning. It tries to fix two big problems: 1) RAG calls breaking the flow of thought (the "tool tax") and 2) multi-agent groups getting bogged down by bad ideas. The main ideas are an "implicit" RAG that injects knowledge when the model seems confused and a "hierarchical" refinement process where solutions are repaired by peers instead of just averaged. The system gets SOTA results on HLE Bio/Chem Gold, hitting 48.3% and beating strong models by a lot , all while using way fewer tokens.

**Strengths:**

The paper offers clever fixes for real problems in agent reasoning. The "Monitor-based RAG" is a great idea; it avoids stopping the reasoning process just to look something up, which clearly saves a lot of compute (53.5% token reduction ). The hierarchical refinement (HSR/QAIR) is also a strong point. Making agents repair each other's work instead of just voting makes a lot of sense and seems to work well based on the ablations. The SOTA results on a tough benchmark like HLE are impressive. I also liked the extra analyses, like the component breakdowns and the look at diversity vs. consensus, which back up the main claims.

**Weaknesses:**

My main concern is that this framework is very complex. There are a lot of moving parts (Monitor, Querier, Injector, HSR, QAIR, etc. ) and it's not obvious how sensitive it is to tuning. The ablation justifies each part, but it's hard to tell how they interact. For instance, does HSR only work well because the Monitor-RAG gives it better solutions to start with, or are those two gains separate? Also, the error analysis is interesting, but the appendix case studies mostly show HSR/QAIR failing to fix the problems. It would be good to see a more direct link between the new components and the specific errors they claim to fix.

**Questions:**

Your system has a lot of parts. Can you say more about how the Monitor-RAG and the HSR/QAIR parts work together? Does the implicit RAG just make the initial solutions better for HSR, or are the improvements from these two modules mostly separate?

You show that reasoning errors are the biggest problem (92.78% of failures ). The Monitor-RAG seems to help with knowledge errors, but how does your system fix pure logical mistakes?

---

> ### Author Response · Authors · 2025-11-20
> **Response to Reviewer h3qQ (1/4)**
>
> Dear Reviewer,
>
> Thank you very much for taking the time to review our work and for your constructive and thoughtful comments.
>
> We appreciate your positive remarks on the novelty of Eigen-1’s implicit retrieval and hierarchical refinement, as well as its token-efficient gains on the challenging HLE Bio/Chem Gold benchmark.
>
> Below, we respond to your main concerns point by point.
>
> &nbsp;
>
> ---
> &nbsp;
>
> # 1. Response to Concern 1: Component complexity and interactions between Monitor-RAG and HSR/QAIR
>
> **Your concern:**
> > “Your system has a lot of parts. Can you say more about how the Monitor-RAG and the HSR/QAIR parts work together? Does the implicit RAG just make the initial solutions better for HSR, or are the improvements from these two modules mostly separate?”
>
> **Our response:**
> Thank you for raising this important question. EIGEN-1 is intentionally designed so that each module plays a distinct, complementary role:
>
> - **Monitor-RAG** detects uncertainty and addresses **knowledge insufficiency** by injecting relevant formulas, definitions, or domain facts.
> - **HSR** repairs **structural reasoning errors**, such as missing intermediate steps, incorrect causal links, or inconsistent logic.
> - **QAIR** provides **quality-aware convergence**, accepting beneficial refinements while filtering out destabilizing or unnecessary edits.
>
> To disentangle these roles, we conducted an additional ablation in which retrieval is **completely disabled**, isolating the impact of HSR+QAIR when they cannot benefit from improved initial solutions from Monitor-RAG:
>
> | Configuration                         | Accuracy (%) | Analysis                                       |
> |--------------------------------------|--------------|-----------------------------------------------|
> | Baseline                             | 25.3%        | Base model performance                        |
> | Baseline + Monitor-RAG               | 43.7%        | Improvement attributable to retrieval alone   |
> | **[New] Baseline + HSR/QAIR**        | **34.5%**    | HSR/QAIR independently correct reasoning errors |
> | Full system (Monitor-RAG + HSR/QAIR) | **48.3%**    | Synergy between retrieval and refinement      |
>
> Even without retrieval, HSR+QAIR improves performance by **+9.2%** over the baseline, demonstrating that their benefits are not solely due to higher-quality initial answers from Monitor-RAG. Instead:
>
> - Monitor-RAG primarily reduces **knowledge errors** (missing formulas, definitions, domain facts).
> - HSR+QAIR primarily reduces **reasoning-chain and methodological errors**, and continues to do so when RAG is off.
> - The full system combines these effects and yields the highest overall accuracy.
>
> We have clarified this interaction and referenced this new ablation in the revised manuscript.

---

> ### Author Response · Authors · 2025-11-20
> **Response to Reviewer h3qQ (2/4)**
>
> # 2. Response to Concern 2: Error analysis
>
> **Your Concern:**
> > “The error analysis is interesting, but the appendix mostly shows cases where HSR/QAIR fail. It would help to see a more direct link between the new components and the specific errors they successfully repair.”
>
> **Our Response:**
> We agree that the previous appendix emphasized failure modes because its original goal was diagnostic.
> To directly address your request, we have now added **five representative success cases** showing how **HSR and QAIR correct Baseline errors**, even when Monitor-RAG is never triggered. These cases span the four dominant categories of reasoning errors:
>
> (1) missing or inconsistent reasoning steps
> (2) arithmetic / numerical inconsistencies
> (3) methodological errors
> (4) peer-evidence consolidation and stabilization
>
> We also include **quantitative statistics** showing how often HSR and QAIR meaningfully improve the solution.
>
> Below we summarize these success cases (each now included in the revised appendix), with the key correction highlighted.
>
> &nbsp;
>
> ---
>
> # 2a. Representative Success Cases (Now Added)
>
> ## **Type 1 — Repairing missing / inconsistent reasoning steps**
>
> ### **Case 1: Pond predator reasoning**
> > **Baseline error:** Prioritizes a smaller warm pond using a temperature heuristic, ignoring the predator cue.
> > **HSR fix:** Reconstructs the causal chain:
> > - young ponds lack predators;
> > - 30-ft pond has **9×** the surface area;
> > - uniform depth removes thermal arguments.
> > ```
> > "Young = predator-free. Larger surface area → more larval habitat.
> > Thus choose the 30-ft, 1-year pond."
> > ```
> > **QAIR:** Verifies predator logic, area calculations, and structural consistency.
> > **Corrected answer:** **C**
>
> ### **Case 2: Predator-vision mismatch**
> > **Baseline error:** Assumes human similarity judgments approximate predator perception.
> > **HSR fix:** Incorporates missing ecological reasoning:
> > - predators may have UV/tetrachromatic vision;
> > - students lack aversive conditioning;
> > - static photos miss motion/background cues.
> > ```
> > "Human rankings generate hypotheses; ecological validity requires
> > predator-vision or multispectral tests."
> > ```
> > **QAIR:** Confirms partial validity conclusions.
> > **Corrected answer:** **Partially valid**
>
> &nbsp;
>
> ## **Type 2 — Resolving numerical/arithmetic inconsistencies**
>
> ### **Case 3: Ozonolysis atom tracking**
> > **Baseline error:** Treats compound 11 as if all atoms survive intact; misassigns nitrogen source.
> > **HSR fix:** Reconstructs the entire fragment lineage:
> > - ozonolysis gives two C₂ fragments;
> > - nitrogen in 10 comes from MeNO₂;
> > - each 14 inherits **two carbons + one oxygen** from 11.
> > ```
> > "... each molecule of 14 inherits one oxygen and two carbons from 11."
> > ```
> > **QAIR:** Verifies inheritance table.
> > **Corrected answer:** **(2, 1, 0)**
>
> &nbsp;
>
> ## **Type 3 — Correcting inappropriate method choices**
>
> ### **Case 4: Jensen-guided allocation**
> > **Baseline error:** Confuses “increasing” with “fair is optimal,” using the wrong mathematical property.
> > **HSR fix:** Applies the correct theoretical tool:
> > - uses **Jensen’s inequality** to evaluate concavity;
> > - constructs counterexamples showing statements 1–3 fail;
> > - leaves only statement 4 as true.
> > ```
> > "Apply Jensen … only statement 4 survives → option D."
> > ```
> > **QAIR:** Checks inequality usage and option mapping.
> > **Corrected answer:** **D**
>
> &nbsp;
>
> ## **Type 4 — Consolidating peer evidence for stable answers**
>
> ### **Case 5: Camera-trap augmentation**
> > **Baseline error:** Focuses on blur-based augmentations despite a major RGB→IR domain mismatch.
> > **Peer rounds:**
> > - Peer A argues for defocus blur (3.0–4.2 m DOF).
> > - Peer B argues for motion blur at 1/500 s.
> > **HSR fix:** Reuses these optical calculations but correctly identifies the **dominant domain shift**:
> > - training = RGB GBIF photos;
> > - deployment = IR grayscale;
> > - colour-noise augmentation enforces shape-based learning.
> > ```
> > "Infrared images are colour-deficient; adding colour noise (J) removes
> > misleading RGB cues and improves deployment accuracy."
> > ```
> > **QAIR:** Confirms blur calculations are acknowledged and that RGB→IR shift drives the final choice.
> > **Corrected answer:** **J**
>
> &nbsp;
>
> ---
>
> # 2b. Quantitative Evidence (from execution logs)
>
> Across the full set of traces:
>
> - **HSR makes a meaningful correction in ~74% of cases**, repairing reasoning-chain, numerical, or methodological errors.
> - **QAIR provides additional refinement in ~63% of cases**, catching subtle inconsistencies and preventing harmful edits.
>
> These statistics confirm that the refinement stack substantially improves solution quality, independently of RAG.
>
>
> The revised appendix now includes both **successes** and **failures**, demonstrating how HSR and QAIR concretely repair baseline errors across multiple reasoning categories.

---

> ### Author Response · Authors · 2025-11-20
> **Appendix of Response to Concern 2 (1/2)**
>
> # Appendix (1/2): Expanded Success Cases Demonstrating HSR and QAIR Corrections
>
> &nbsp;
>
> ---
> &nbsp;
>
> To complement the summary provided in **Response to Concern 2**, we include here *extended versions* of five representative success cases.
> These examples illustrate how **HSR** and **QAIR** correct Baseline errors across the four dominant categories of reasoning failures:
>
> 1. Missing or inconsistent reasoning steps
> 2. Arithmetic / numerical inconsistencies
> 3. Methodological errors
> 4. Peer-evidence consolidation and stabilization
>
> Each case includes:
> - the Baseline error,
> - the corrected HSR chain,
> - relevant peer evidence (when applicable), and
> - QAIR verification logs.
>
> These full examples are now included in the revised Appendix of the manuscript.
>
> &nbsp;
>
> ---
> &nbsp;
>
> # **A.1 Case 1 — Pond Predator Reasoning (Missing Steps Repair)**
>
> **HLE Question.**
> “All ponds are cement boxes (2 ft deep; 10 ft or 30 ft square). Half were built one year ago and half five years ago. Older ponds now host established insect communities. With a single emergence trap, choose the pond posing the greatest medical threat.”
>
> **Baseline Answer (Incorrect).**
> “Older ponds have more insects, so they host more mosquitoes. Cooling is slower in small ponds, hence the 10-ft 1-year pond is worst.”
>
> ➡ **Baseline mistake:**
> Uses a *temperature heuristic* irrelevant to cement tanks and ignores the explicit *predator cue* (“older ponds have insect communities”).
>
> ---
>
> ## **HSR Correction (Successful)**
>
> HSR reconstructs the causal chain:
>
> - Young ponds lack predators.
> - Larger ponds provide vastly more larval substrate:
>   - 30 ft pond = **900 ft²**
>   - 10 ft pond = **100 ft²**
> - Identical depth + cement walls remove all thermal arguments.
>
> ```
> "Young = predator-free. Larger surface area → more egg-laying substrate.
> The 30-ft pond provides 9× the larval habitat, so choose the 30-ft,
> 1-year pond."
> ```
>
> ## **QAIR Verification**
> ```
> Performed:
> - Predator-control logic ................ PASS
> - Surface-area calculation (900 vs 100) .. PASS
> - Temperature-heuristic audit ............ PASS
>
> "... choose the young 30-ft pond; predators colonize older ponds,
> and 9× surface area increases larval habitat. <answer>\boxed{C}</answer>"
> ```
>
> **Corrected Answer:** **C**
>
> ---
>
> # **A.2 Case 2 — Predator-Vision Mismatch (Missing Ecology Reasoning)**
>
> **HLE Question.**
> “Twenty undergraduates rank visual similarity of *Bombus* species from photos; researcher clusters mimicry syndromes based on that consensus. Is this valid for the ecological function of mimicry?”
>
> **Baseline Answer (Incorrect).**
> “Human perception is close to predator perception, so the method is acceptable.”
>
> ➡ **Baseline mistake:**
> Assumes humans ≈ predators; ignores UV/tetrachromatic vision and learning-based predator behavior.
>
> ---
>
> ## **HSR Correction (Successful)**
>
> HSR incorporates missing ecological reasoning:
>
> - Predators see UV or have different color channels.
> - Predator avoidance is learned; students lack conditioning.
> - Static photos omit motion/background cues relevant to aposematic displays.
>
> ```
> "Undergraduate rankings can seed hypotheses, but ecological validity
> requires predator-vision models or behaviour assays. Thus only partially valid."
> ```
>
> ## **QAIR Verification**
> ```
> Performed:
> - Partial-utility acknowledgement .... PASS
> - Need for predator-vision evidence .. PASS
>
> "Human rankings generate hypotheses; ecological validity still
> requires predator-vision tests. <answer>Partially valid</answer>"
> ```
>
> **Corrected Answer:** **Partially valid**

---

> ### Author Response · Authors · 2025-11-20
> **Appendix of Response to Concern 2 (1/2)**
>
> # Appendix (2/2): Expanded Success Cases Demonstrating HSR and QAIR Corrections
>
> &nbsp;
>
> ---
> &nbsp;
>
> # **A.3 Case 3 — Ozonolysis Atom Tracking (Arithmetic / Numerical Repair)**
>
> --
>
>
> **HLE Question.**
> Track how many carbons/oxygens from compound 11 survive into compound 14 and multifidene (1), and how many nitrogens from 7 appear in 10.
>
> **Baseline Error.**
> Treats 11 as if all atoms persist; assumes nitrogen from 7 back-propagates into 10.
>
> ---
>
> ## **HSR Correction (Successful)**
>
> HSR reconstructs correct fragment lineage:
>
> - Ozonolysis → two **C₂** fragments.
> - Only one TES-bearing fragment proceeds through 12 → 10 → 13 → 14 → 1.
> - Nitrogen in 10 comes from **MeNO₂**, not 7.
> - Each 14 receives:
>   - **2 carbons** from 11
>   - **1 oxygen** from 11
>   - **0 nitrogens** from 7
>
> ```
> "After O3 cleavage, only one TES fragment continues. Nitrogen in 10 comes
> from MeNO2. Each 14 inherits two carbons and one oxygen from 11."
> ```
>
> ## **QAIR Verification**
> ```
> "Fragment sketch added. Correct tally:
> 2 carbons, 1 oxygen, 0 nitrogens. <answer>2, 1, 0</answer>"
> ```
>
> **Corrected Answer:** **(2, 1, 0)**
>
> ---
>
> # **A.4 Case 4 — Jensen-Guided Allocation (Methodological Correction)**
>
> **HLE Question.**
> Given a strictly concave/convex function \(s(r_i)\) and total resources \(R\), identify which statements about fair vs extreme allocation are true.
>
> Correct answer: **D (only statement 4)**.
>
> **Baseline Error.**
> Applies monotonicity heuristics (“strictly increasing → fair is optimal”), ignoring curvature.
>
> ---
>
> ## **HSR Correction (Successful)**
>
> HSR applies correct functional analysis:
>
> - Uses **Jensen’s inequality** to identify when equal splitting is optimal.
> - Provides explicit counterexamples showing statements 1–3 fail.
> - Retains only statement 4.
>
> ```
> "Apply Jensen: for concave s, sum s(r_i) ≤ n·s(mean). Construct
> counterexamples for 1–3. Only statement 4 holds → option D."
> ```
>
> ### QAIR Verification
> ```
> Performed:
> - Jensen inequality reference ........ PASS
> - Counterexample construction ........ PASS
> - Option mapping ..................... PASS
>
> "Only statement 4 survives → \boxed{D}."
> ```
>
> **Corrected Answer:** **D**
>
> ---
>
> # **A.5 Case 5 — Camera-Trap Augmentation (Peer-Evidence Integration)**
>
> **HLE Question.**
> Camera-trap deployment uses IR grayscale images; training uses colorful GBIF photos. Which augmentation is most important?
>
> **Baseline Error.**
> Chooses blur-based augmentations (Gaussian/motion) and ignores the RGB→IR domain shift.
>
> ---
>
> ## **Peer Evidence (Cross-Agent Round)**
>
> - **Peer A:** Gaussian blur due to DOF
>   > DOF ≈ 3.0–4.2 m; defocus blur expected at f/4.
> - **Peer B:** Motion blur due to animal movement even at 1/500 s.
>
> ---
>
> ## **HSR Correction (Successful)**
>
> HSR integrates peer arguments but identifies the *dominant mismatch*:
>
> - Training: RGB color images
> - Deployment: IR monochrome → **massive color-channel mismatch**
> - Optical blur differences are smaller than spectral differences
>
> ```
> "Infrared images are colour-deficient; augmentations that disrupt RGB
> patterns—such as colour noise (J)—force reliance on shape, not hue.
> This addresses the dominant training→deployment shift."
> ```
>
> ---
>
> ## **QAIR Verification**
> ```
> Checks:
> - DOF and motion-blur calculations acknowledged ........ PASS
> - RGB→IR domain-shift identification .................... PASS
> - Final augmentation justified by dominant mismatch ..... PASS
>
> "Colour noise (J) corrects the major RGB→IR discrepancy."
> ```
>
> **Corrected Answer:** **J**
>
> ---
>
> # **A.6 Quantitative Summary**
>
> Across all execution traces:
>
> - **HSR provides meaningful corrections in ~74% of cases.**
> - **QAIR provides additional improvements in ~63% of cases**, preventing harmful refinements and stabilizing valid reasoning.
>
> ---
>
>
> These expanded examples demonstrate how **HSR** and **QAIR** concretely correct reasoning-chain, arithmetic, methodological, and peer-integration errors.
> All examples are now included in the revised appendix to provide full transparency of the correction process.

---

> ### Author Response · Authors · 2025-11-20
> **Response to Reviewer h3qQ (3/4)**
>
> # 3. Response to Concern 3: Linking components to specific error types
>
> **Your Concern:**
> > “It would be good to see a more direct link between the new components and the specific errors they claim to fix.”
>
> **Our Response:**
> We agree that making these connections explicit improves the interpretability of the framework. In the revised appendix, we now include:
> (a) a component–error mapping table,
> (b) quantitative statistics extracted from execution logs, and
> (c) qualitative examples (referenced in our Response to Concern 2).
>
> Together, these additions clarify how each module mitigates distinct types of reasoning errors.
>
> ## **a) Component–Error Mapping**
>
> To make the role of each module precise, we now provide an explicit mapping between error categories and the components responsible for addressing them:
>
> | Error Type                               | Monitor-RAG         | HSR                  | QAIR                  |
> |-------------------------------------------|----------------------|----------------------|------------------------|
> | Knowledge insufficiency                   | **Primary fix**      | —                    | —                      |
> | Reasoning-chain errors                    | Flags uncertainty    | **Primary fix**      | Secondary consistency check |
> | Numerical / arithmetic errors             | —                    | Targeted correction  | Re-verification        |
> | Methodological errors                     | Sometimes helpful    | **Strong fix**       | Filter for incorrect revisions |
> | Overconfidence / incomplete justification | —                    | —                    | **Final quality check** |
>
> This table is now included in the updated appendix.
>
> ## **b) Quantitative Evidence**
>
> Using the full execution trace, we further quantified the effect of each module:
>
> - **Monitor-RAG** reduces knowledge-insufficiency errors in monitored segments by approximately **45%**.
> - **HSR** corrects reasoning-chain and method-selection errors in roughly **33%** of applicable cases.
> - **QAIR** resolves residual logical-completeness issues after HSR in approximately **18%** of cases.
>
> These additions clarify the division of labor within EIGEN-1:
> **Monitor-RAG addresses factual gaps, HSR repairs structural reasoning errors, and QAIR provides the final quality-control layer.**

---

> > ### Author Response · Authors · 2025-11-20
> > **Response to Reviewer h3qQ (4/4)**
> >
> > # 4. Response to Concern 4: How the system repairs *pure logical mistakes*
> >
> > **Your Concern:**
> > > “You show that reasoning errors are the biggest problem (92.78% of failures).
> > > The Monitor-RAG seems to help with knowledge errors, but how does your system fix pure logical mistakes?”
> >
> > **Our Response:**
> > Thank you for raising this important clarification. As the reviewer notes, the vast majority of baseline failures (92.78%) arise from **reasoning errors rather than missing knowledge**. In EIGEN-1, these errors are repaired not by Monitor-RAG, but by **HSR** (Hierarchical Solution Refinement) and **QAIR** (Quality-Aware Iterative Revision).
> > To directly address the reviewer’s question, we added multiple case studies showing that:
> >
> > 1. **HSR reconstructs incomplete or partially incorrect causal chains**,
> > 2. **HSR corrects invalid inferences even when all facts are already present**,
> > 3. **QAIR detects subtle inconsistencies introduced during refinement**, and
> > 4. **These repairs occur even when Monitor-RAG is never triggered** (i.e., they are truly “pure logic” fixes).
> >
> > Below, we provide the representative examples now included in the revised appendix.
> >
> > ---
> >
> > ## **a) HSR reconstructs incomplete or partially incorrect reasoning chains**
> >
> > > ### **Case 1: Chain-completion error**
> > > **HLE question:** Antiseizure therapy → cognitive decline → severe blistering (mechanism asked).
> > > **Baseline error:**
> > > Names lamotrigine oxidation, but omits (i) valproate’s UGT inhibition, (ii) forced diversion to P450 oxidation, and (iii) downstream immune cascade (hapten formation → SJS/TEN).
> > >
> > > **HSR repair:**
> > > - Identifies valproate-mediated UGT blockade.
> > > - Reconstructs diversion to CYP2A6/2D6 oxidation forming arene-oxide.
> > > - Connects reactive metabolite to keratinocyte haptenization → SJS/TEN initiation.
> > > ```text
> > > “Valproate blocks glucuronidation, diverting lamotrigine into P450-mediated
> > > arene-oxide formation. When detox pathways saturate, the arene-oxide
> > > haptenizes keratinocyte proteins and triggers the SJS/TEN cascade.”
> > > ```
> > > **QAIR:** Confirms chain completeness, enzyme specificity, and removes digressions.
> > > **Final answer:** Correct biochemical initiating event.
> >
> > ---
> >
> > ## **b) HSR corrects invalid logical inferences even when all facts are known**
> >
> > > ### **Case 2: Logical inference error (heritability H² = 0.7)**
> > > **Baseline error:**
> > > Concludes that west vs. east mean phenotypes must be identical because the population is infinite and genetically uniform—incorrectly equating **genetic homogeneity** with **phenotypic homogeneity**, ignoring environmental variance (30%).
> > >
> > > **HSR repair:**
> > > - Separates what is *given* (no selection, mutation, or gene flow) from what must be *derived*.
> > > - Points out H² < 1 allows environmental spatial variation → phenotypic means may differ.
> > > - Correctly locks statements 1 and 3 as the only necessarily true ones.
> > > ```text
> > > “H² = 0.7 still leaves 30% environmental variance; west–east phenotypic
> > > differences may arise even in an infinite HW population. Only statements 1
> > > and 3 are forced → option F.”
> > > ```
> > > **QAIR:** Confirms proper use of heritability and speciation logic.
> >
> > ---
> >
> > ## **c) QAIR catches subtle consistency errors introduced during refinement**
> >
> > > ### **Case 3: Parsimony tree (Fitch evaluation)**
> > > **Baseline error:**
> > > Asserts “Tree C has 6 changes” without showing trait matrix or per-edge counts.
> > >
> > > **HSR attempt:**
> > > Adds matrix + reasoning, but mis-codes traits and omits Fitch step-by-step counts.
> > >
> > > **QAIR repair:**
> > > ```text
> > > Performed:
> > > - Consistency audit ............. FAIL
> > > - Character-coding check ........ FAIL
> > > - Per-tree step count audit ..... FAIL
> > > Action required: provide corrected matrix + explicit Fitch counts.
> > > ```
> > > Only after HSR adds the corrected table and per-topology step counts does QAIR accept the solution.
> > > **Final outcome:** Verified 6-change minimum for Tree C.
> >
> > ---
> >
> > ## **d) Pure reasoning repairs occur without Monitor-RAG**
> >
> > Across these examples—and in many of the additional cases added to the appendix—**no retrieval is triggered**.
> > HSR rewrites the reasoning chain, and QAIR ensures consistency and correctness, showing that:
> >
> > - *Pure logical mistakes*
> > - *Incomplete chains*
> > - *Incorrect inference rules*
> > - *Subtle methodological/logical inconsistencies*
> >
> > are handled by the **HSR + QAIR refinement stack**, independent of RAG.
> >
> > ---
> >
> > **Conclusion:**
> > While Monitor-RAG targets knowledge insufficiency, the dominant error class—**reasoning errors (92.78%)**—is handled by HSR and QAIR.
> > We now make this role separation explicit through:
> >
> > - retrieval-disabled ablations,
> > - new reasoning-only success cases,
> > - and a component–error mapping table.

---

### Author Response · Authors · 2025-12-03
**Author Summary for AC**

Dear Area Chair,

Thank you for your time throughout the review process. We appreciate the constructive feedback from all reviewers (h3qW, iFym, nD2g). We have conducted substantial additional experiments and incorporated all updates into the revised PDF on OpenReview.

During the discussion phase, we engaged actively with reviewers and provided targeted new analyses addressing their main concerns:

- **Reviewer h3qW (Score: 6)** – Clarified the distinct roles of Monitor-RAG vs. HSR/QAIR using new ablations and detailed case studies, showing that logical repairs occur even without retrieval.
- **Reviewer iFym (Score: 6)** – Evaluated robustness across prompt variants, base-model changes (o4-mini), and evaluator settings, confirming stability and model-agnostic behavior.
- **Reviewer nD2g (Score: 6)** – Conducted cross-corpus experiments with a Generic Mixed-Domain RAG and performed a manual leakage audit, verifying that improvements come from the architecture rather than domain-specific alignment.

Below is a concise summary of the main technical updates:

---

### 1. Component Interaction & Logical Error Repair (Reviewer h3qW)

- **Reasoning-only ablation:** Added a *no-retrieval* configuration (Baseline + HSR/QAIR), yielding **+9.2%**, isolating refinement gains from retrieval.
- **Success-case analysis:** Added five representative cases and quantitative data showing HSR/QAIR fix reasoning-chain, arithmetic, and methodological errors without Monitor-RAG.
- **Component–error mapping:** Included a table linking each error type to the module responsible (Monitor-RAG, HSR, QAIR).

---

### 2. Robustness & Generalization (Reviewer iFym)

- **Model-agnostic validation:** Re-ran EIGEN-1 on **o4-mini**, achieving **+21.5%** improvement, similar to DeepSeek-V3.1’s **+23.0%**.
- **Prompt sensitivity:** Tested paraphrased and concise prompts; accuracy stayed within –1.4% to +0.6%.
- **Evaluator reliability:** Human validation shows **96.7%** agreement with QAIR decisions and **100%** agreement with the final-answer judge, mitigating evaluator–solver coupling concerns.

---

### 3. RAG Validity & Domain Alignment (Reviewer nD2g)

- **Generic Mixed-Domain RAG:** Using a 14k-paper, 148-field corpus, EIGEN-1 achieves **45.6%** (+20.3% over baseline), showing gains are mechanism-driven.
- **Reasoning-dominant gains:**
  - Baseline: 25.3%
  - Baseline + HSR/QAIR: 34.5%
  - Generic RAG: 45.6%
  - Bio/Chem RAG: 48.3%
  Domain knowledge adds a modest incremental gain.
- **Leakage audit & transparency:** Manual audit found **0** cases of answer leakage. We will open-source the full RAG corpus and index.

---

These updates address all reviewer concerns and significantly strengthen the manuscript.

Best regards,

Authors of Paper 18203

---

### Meta-Review · Area_Chair_fvMY · 2026-01-04

**Summary:**

The paper presents a new multi-agent system for scientific reasoning. It has many valuable components such as monitor-based RAG, which interleaves reasoning and retrieval. Another component is hierarchical refinement -- agents repairing each others' work. The results on tough benchmarks (HLE Bio/Chem) were considered impressive by reviewers.

Reasons to accept: novel components combined in a unified system, detailed ablations and error analyses, impressive results, easy to deploy

Reasons to reject: no results on abstract reasoning benchmarks suxh as math/physics.

**Reviewer Concerns:**

Lots of concerns were raised by reviewers. They included detailed error analyses, additional ablations, additional expts with other base models, robustness to prompt variations, dependence of experiments on domain-specific corpora, etc.

The authors provided detailed notes -- new samples for explaining errors and strengths of the model... plus, many additional expts answering questions. The only question they did not answer is whether their approach is suitable for physics/math abstract reasoning.

While this lack of experiment does diminish some excitement in the paper, even if the work is valuable on reasoning tasks that require a fair bit of knowledge, I am ok... since those are valuable problems as well.

**Reviewer Scores:**

I expect that two of the 6s would have moved to 7s.

---

### Decision · Program_Chairs · 2026-01-26

Accept (Poster)